# Probing complex geophysical geometries with chattering dust

Laura J. Pyrak-Nolte [1,2,3,6 ✉], William Braverman[4,6], Nicholas J. Nolte[5,6], Alan J. Wright[1,6] & David D. Nolte[1,6]

The modern energy economy and environmental infrastructure rely on the flow of fluids through fractures in rock. Yet this flow cannot be imaged directly because rocks are opaque to most probes. Here we apply chattering dust, or chemically reactive grains of sucrose containing pockets of pressurized carbon dioxide, to study rock fractures. As a dust grain dissolves, the pockets burst and emit acoustic signals that are detected by distributed sets of external ultrasonic sensors that track the dust movement through fracture systems. The dust particles travel through locally varying fracture apertures with varying speeds and provide information about internal fracture geometry, flow paths and bottlenecks. Chattering dust particles have an advantage over chemical sensors because they do not need to be collected, and over passive tracers because the chattering dust delineates the transport path. The current laboratory work has potential to scale up to near-borehole applications in the field.

[1] Purdue University, Department of Physics & Astronomy, West Lafayette, IN 47907, USA. [2] Purdue University, Department of Earth, Atmospheric and Planetary Sciences, West Lafayette, IN 47907, USA. [3] Purdue University, Lyle School of Civil Engineering, West Lafayette 47907 IN, USA. [4] Louisiana State University, Department Physics & Astronomy, Baton Rouge, LA 70803, USA. [5] University of California, Berkeley, Berkeley, CA 94720, USA. [6] These authors contributed equally: Laura J. Pyrak-Nolte, William Braverman, Nicholas J. Nolte, Alan J. Wright, David D. Nolte. ✉email: ljpn@purdue.edu

A wide range of subsurface activities, including resource extraction from geothermal energy systems, sequestration of greenhouse gases, and the protection of potable aquifers, rely on the transport of fluids through the Earth's crust. This crucial transport occurs primarily along fractures or through connected fracture networks, in which the flow rates are intimately linked to the geometry and spatial distribution of fracture apertures[1–7]. However, the remote assessment of fracture apertures is problematic because rocks are opaque to most probes. One common method used in the field to infer fracture properties relies on passive geophysical signals generated by local abrupt failures (microseismicity or acoustic emissions) from the fractures themselves, as they respond to changing stresses. However, these microseismic approaches are limited because nonselective events can occur anywhere within the subsurface, even away from fractures of interest[8,9], non-seismic events may give no measurable signal of fracture evolution (e.g., ref. [10]), and interpretation of events provide little or no information on fracture connectivity or fracture aperture[11].

There was a push to develop nano- or micro-sensors that could be transported and distributed into a subsurface formation to report back information on rock structure, lithology, fractures, and pore connectivity among other properties. This effort was partially driven by the development of biophotonic sensors, such as functionalized photonic crystals, made from porous silicon, known as "smart dust"[12], as well as photonic devices made of polymers or silica matrix materials known as "PEBBLES"[13,14] that had a wide range of potential applications outside of geophysics. These functional sensors encouraged the development of subsurface sensing that relied on reservoir reporters and smart tracers[15–20] to measure proppants in fractures, or to detect the presence of hydrocarbons. However, these subsurface sensor-based methods require the sensors to flow into and to be recovered out of a subsurface system to enable readout of the sensors. While the recovery of such smart sensors from different boreholes demonstrate that two boreholes are connected, such sensors provide no information on the topology of the flow path between boreholes.

We have developed a method using so-called chattering dust to overcome these limitations through the transport of time-release microseismic grains through fracture flow paths. This approach combines tracer testing with remote geophysical monitoring to illuminate the connected flow paths in fractures and to infer variations in fracture aperture, as the dust is transported along flow paths. In this paper, we demonstrate seismic location and tracking of chattering dust grains as they transport through fractures and through fracture intersections, where complex grain trajectories are observed, caused by eddy currents. The current laboratory work has potential to scale-up to near-borehole and civil infrastructure applications in the field.

## Results

**Chattering dust**. Chattering dust refers to chemically reactive millimeter-scale grains that emit a sequence of acoustic shocks as the grains dissolve. The acoustic shocks are detected using acoustic emission wave sensors and are located using triangulation (see Supplementary information). Chattering dust is composed of commercially available reactive grains made of sucrose containing pockets of pressurized carbon dioxide (4.1 MPa; Figs. 1 and 2). As the coating dissolves, the compressed gas is explosively released, yielding acoustic emissions (e.g., Supplementary Figs. 4 and 6c). X-ray microscopy shows that the dust grains contain spherical bubbles of pressurized gas that range in size from 4.5 to 270 μm (Figs. 1 and 2). For instance, a single reactive dust grain contains a distribution of bubble sizes

throughout the bulk of the sample. The volume probability distribution functions for the bubbles compared to the distribution functions for the amplitudes of the events show relatively similar functional forms (Fig. 2). The lower cutoffs of the two distributions are not equivalent, and the energy distribution extends to lower relative values. In Fig. 2b, e, an estimate of the energy released can be obtained from the amplitude, as the energy of a signal is proportional to the amplitude square. The dust is denser than water, varies in size from 7.34 to 25.27 mm$^3$ (also see Supplementary Table 2 and Supplementary Table 5), and a single intact dust grain emits 100's–1000's of acoustic event signals (Fig. 2 and Supplementary Table 4) over a 3–6 min period (Fig. 3b, e). As it dissolves (at a rate of ~0.115 mm$^2$/s ± 0.014 mm$^2$/s; see Fig. 3), the average time between recorded emissions is ~30 ms and the average amplitudes of the events remain relatively constant during dissolution (Fig. 3c, f). The decay in the number of events follows the same trend as the decay in the area of the particle (Fig. 3b, e). The functional form of the dissolution rate (change in area per time, see Supplementary Note 4 for details) of the grains differs between particles that exhibited disaggregation (Fig. 3d sample DT14) compared to those that did not (Fig. 3a sample DT4). When a dust grain did not disaggregate, the area and number of hits decreased approximately linearly in time. For particles that disaggregated (DT13–DT15), the area and number of hits decreased approximately exponentially. Although the number of events decay, the average amplitude of the events is relatively constant (Fig. 3c, f), which ensures detectability over the duration of the existence of a particle. Single dust grains with a larger initial area exhibited a longer duration of acoustic emissions (Supplementary Fig. 6a).

**Seeding a fracture with chattering dust**. To explore the feasibility of using chattering dust to detect and track an imbibition front, as water imbibes into a fracture, we pre-seeded a dry fracture with a sparse distribution of individual grains of chattering dust (Fig. 4b). Because fracture aperture is one of the controls on how a fluid invades a fracture, a nonuniform invading front occurs during immiscible fluid displacement[21–23] in a variable-aperture fracture or pore network. From the Laplace–Young's equation, the aperture penetrated by a fluid is inversely proportional to the capillary pressure between two immiscible phases[23,24]. In our lab, a transparent cast of a variable-aperture fracture was seeded with chattering dust (Fig. 4b) to track an invading fluid front, as water imbibed into an air-filled fracture. The fracture was a transparent replica of a natural fracture from the Stripa Mine, Sweden, with dimensions of 0.219 m × 0.307 m and an average thickness of 37.28 mm that was previously characterized by Su et al.[25] to have a mean fracture aperture of 160 μm ± 110 μm, when compressed together between two flat Lucite plates and bolted together between aluminum flanges. Here, the same fracture was used but in an unconfined-unsealed condition (see experimental details in the Supplementary information). The spatial distribution of fracture apertures is shown in Fig. 4a with the color indicating the size of the aperture.

The sequential position of the invading front was tracked over the duration of the experiment, using a digital camera and acoustic location that uses acoustic emission signals from the chattering dust (Fig. 4c). Though water was invaded across the entire bottom edge of the fracture plane (position = y = 0), the fluid front did not invade uniformly because of the variable aperture across the fracture. Based on the aperture map from reference[25], the center of the fracture has small-aperture voids (red-black regions in Fig. 4a) with larger-aperture voids to the left and right (blue-green regions in Fig. 4a). The time-lapse acoustic data show that first emission (light green stars in Fig. 4c) did not

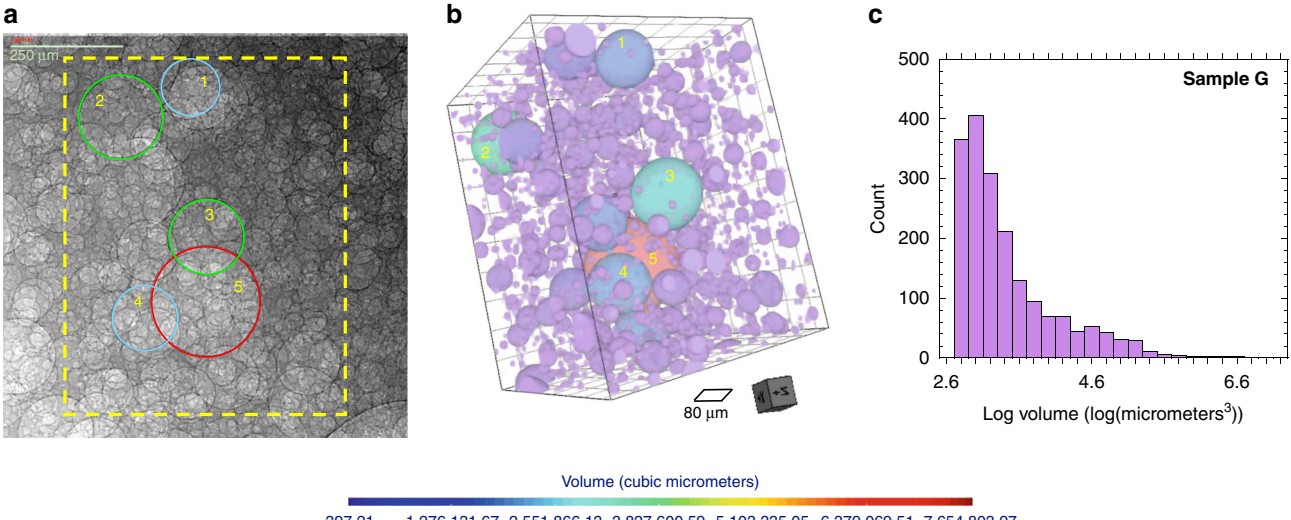

**Fig. 1 Bubble geometry in a dust grain. a** 2D projection (dashed yellow outlines reconstructed region in **b**). **b** 3D reconstruction of a subregion (622.34 × 797.43 µm × 600.57 µm$^3$) of a single grain taken with a 20× objective on a 3D X-ray microscope showing bubbles with radii greater than five pixels. The color scale in **b** represents bubble volume. The red bubble in **b** is the bubble circled in red in **a**. The diameter of the red circle in **a** is 243 µm. The pixel resolution is given in Supplementary Table 1. **c** Probability distribution of the log of bubble volume for bubbles shown in **b**.

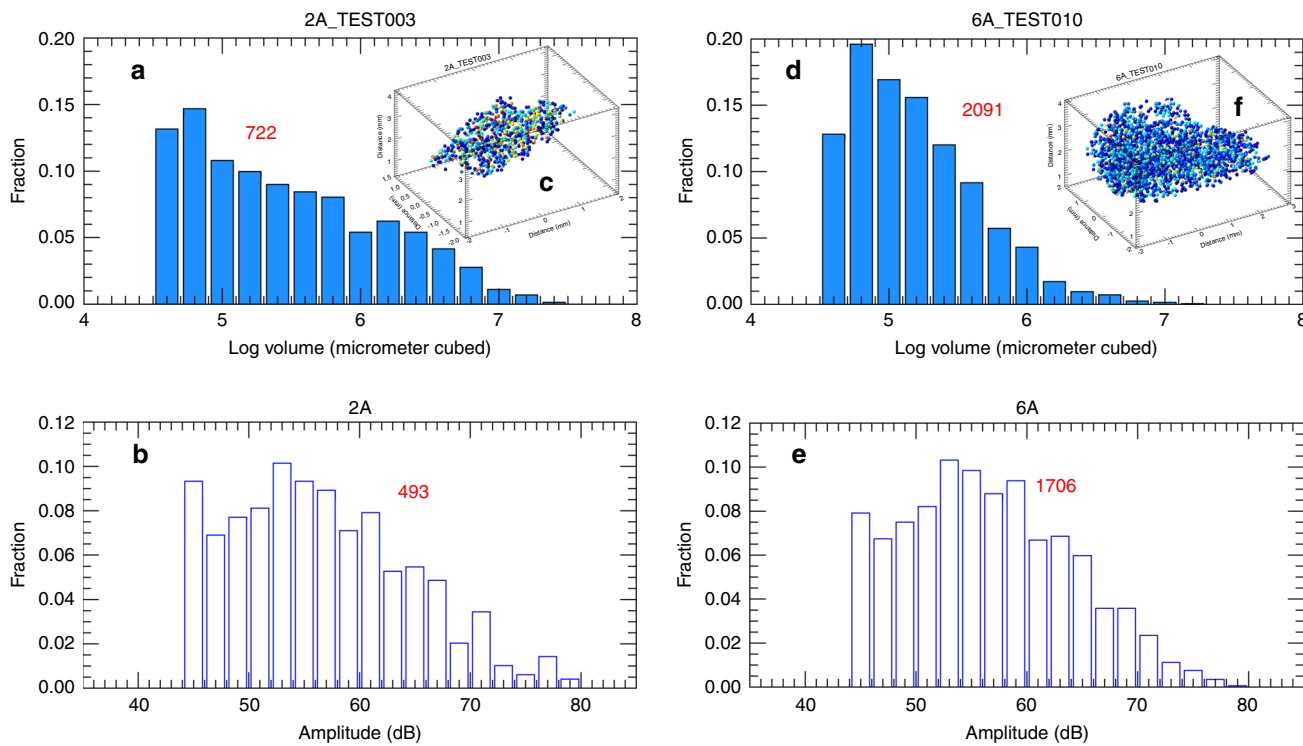

**Fig. 2 Bubble and acoustic amplitude distributions. a**, **d** Probability distribution of bubble volume (in $\log_{10}$) for single-grain samples CD2A and CD6A (see Supplementary Table 1). **b**, **e** Distribution of the logarithm of energy generated by the bubbles in the grain (number of recorded events shown in red and based on recorded events from one of the sensors). **c**, **f** 3D visualization of the center of mass of the bubbles within each single grain (number in red is number of bubbles) from X-ray imaging.

occur until 10 min after the fluid started imbibing (light green contour in Fig. 4c). A finger of small apertures in the mid-fracture plane ($x \sim 0.1$ between $0.1\,\text{m} < y < 0.15\,\text{m}$) resulted in a delay of the fluid front movement, as indicated by the dip in the time-lapse acoustic map near a horizontal position of 0.13 mm. This demonstrates that the chattering dust does not emit until water reaches a dust grain and dissolution begins. Some of the locations of the emissions are near the original placement of the dust, while others are not. During setup of the fracture, the dust is crushed when the two surfaces are registered, resulting in multiple particles (e.g., multiple stars near sensors 1, 5, and 6 in Fig. 4b) that can potentially move with the invading front. The acoustic locations agree with the visual location of the fluid front to within ±5 mm. This agreement illustrates the potential to use chattering dust to track fluid front movement through fractures in opaque material (i.e., rock) at the laboratory scale and potentially at the

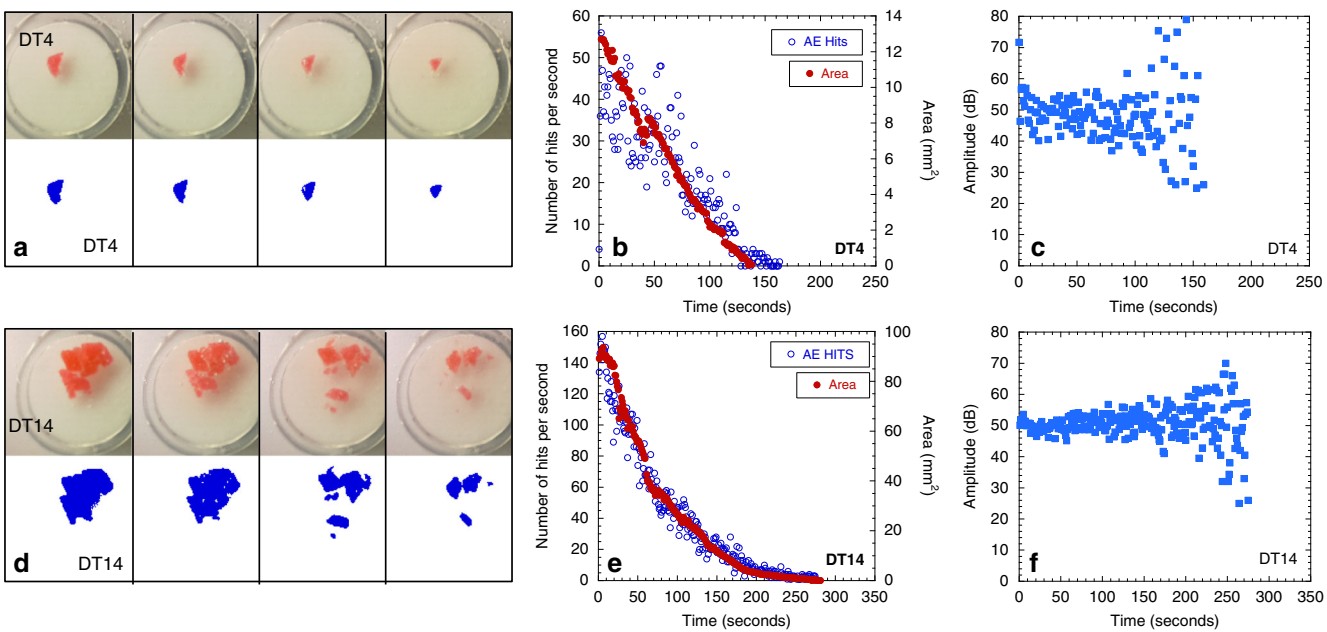

**Fig. 3 Dissolution of a dust grain. a, d** A digital image showing a dust grain sitting in a petri dish over an AE sensor, and segmented images of the dust for sample DT4, which did not disaggregate and sample DT14, which disaggregated. **b, e** Change in grain size area and number of hits per second as a function of time for samples DT4 and DT14. **c, f** Average signal amplitude as a function time for samples DT4 and DT14.

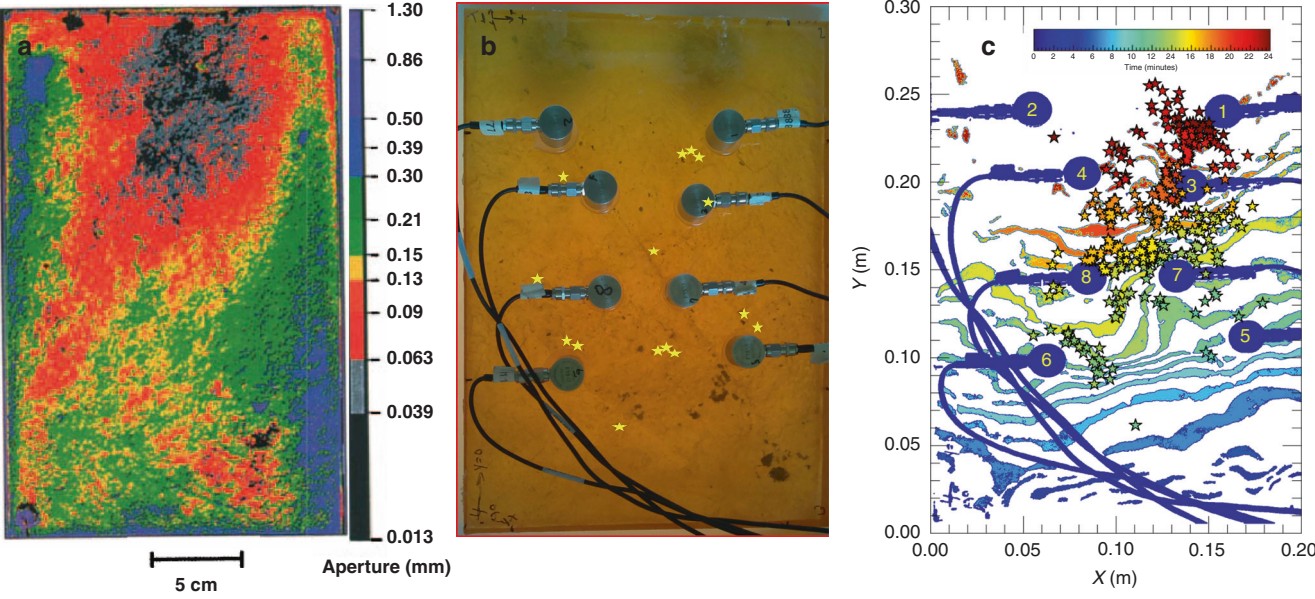

**Fig. 4 Acoustic emissions during fluid invasion in a dust-seeded fracture. a** The aperture map adapted from Su et al.[25] with the color representing the size of the aperture in mm. **b** A digital image of the fracture sample with AE sensors. Yellow stars indicate the location of individual dust grains. **c** Time-lapse of acoustic emission locations from an imbibing fluid front ($+y$-direction is the imbibition direction) in a fracture pre-seeded with chattering dust. The sensor locations are shown in blue with the sensor number in yellow. The color scale represents time for both the invasion contours and the AE events (stars). The stars indicate the triangulated locations of the individual events. Each invasion contour represents the region the fluid invaded during a 20 s window with 60 s between the contours.

near-borehole scale. The location uncertainty is caused by the water–air interface, the fracture roughness, and the high-angle detection geometry (see Supplementary Note 4).

**Chattering dust settling under gravity.** In contrast to pre-seeding a fracture, laboratory demonstrations of dust transport through fractures can use gravitational settling or active flow conditions. From Stokes' equation, the settling velocity for a sphere under the influence of gravity is proportional to the square

of the diameter of the sphere[26,27]. If the sphere is placed between parallel plates (i.e., uniform-aperture fracture), the sphere experiences normal and tangential forces from the wall for which the characteristic length scale is the ratio of the diameter of the particle to the distance from the wall[28]. As the aperture of a fracture increases, the normal and tangential forces decrease and the speed of the sphere approaches that for open tank (Fig. 5b). For chattering dust, this relationship between sphere diameter and fracture aperture enables us to infer changes in fracture

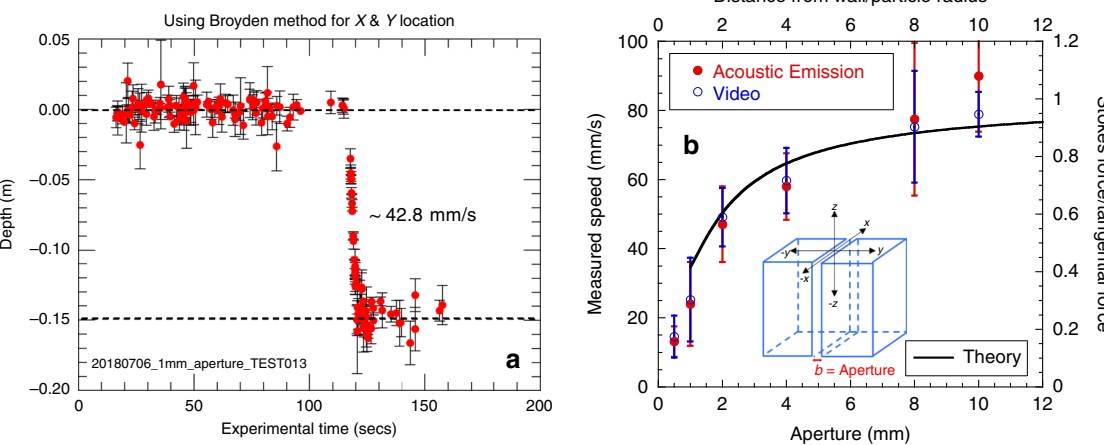

**Fig. 5 Descent speed. a** Depth of emitting grain of a dust in a 1 mm uniform-aperture fracture as a function of time. **b** Average dust speed as a function of fracture aperture from video image processing and acoustic emission localization (see Supplementary Note 6 for approach). Tangential drag decreases as the aperture of the fracture increases. Also shown is the theoretical prediction of the Stokes force normalized by the force from the walls, as a function of the ratio of distance of a sphere from the wall divided by the sphere radius, assuming the sphere is centered between the two fracture walls.

apertures, by tracking the location of the source as a function of time. This behavior was demonstrated in experiments for a range of uniform apertures (0.5–10 mm), as well as through variable apertures in transparent synthetic fractures that were used to directly image the dust location, while simultaneously recording acoustic emissions from the chattering dust. Details of the the experiments and the localization methods are given in the Supplementary Note 6.

*Uniform-aperture fracture*: To test the acoustic localization of chattering dust grains falling under gravity, dust grains were released individually into synthetic transparent fractures. Initially a grain would rest on the air–water interface, providing a reference point for the initial position (duration 110 s as observed in Fig. 5a, and also see Supplementary Fig. 9a). Then the emitting dust grain would submerge and descend into the fracture, eventually coming to rest on the bottom of the fracture. Figure 5a shows the located depth of a dust grain as a function of time, as it settled under the influence of gravity in a 1 mm uniform-aperture fracture. The speed of the grain determined by remote acoustic sensing was 43 mm s⁻¹ compared to 41 mm s⁻¹ extracted from the video that monitored the falling grain. The average dust grain descent speed increases with increasing aperture (Fig. 5b). The increase in speed as a function of fracture size follows classic behavior for a sphere falling between two walls[29] (Fig. 5b).

Monitoring changes in velocity as a chemically reactive source moves through a fracture is indicative of variations in fracture aperture, which controls fluid flow. This was tested on fractures with uniform apertures that ranged between 0.5 and 10 mm. As the fracture aperture increases, the speed of the grain falling under gravity increases. This increase was confirmed from analysis of video images of emitting dust grains falling under gravity in each of the fractures, and represents the speed averaged over 7–10 runs per aperture size. Figure 5b also shows the Stokes force ($F_S = 6\pi\mu a v_p$, where $\mu$ is the fluid viscosity, a is the particle radius, and $v_p$ is the particle velocity) normalized by the tangential force from the fracture walls ($F_T = F_S/C$ with $C = 1 - A\frac{a}{d} + B\left(\frac{a}{d}\right)^2$, where $d$ is the distance from the particle to the wall, and $A$ and $B$ are constants[28]), as a function of distance from the wall for a sphere between two infinite parallel plates. For two parallel walls, the upper horizontal axis represents the distance between the two walls divided by the particle radius. The drag from the wall decreases with increasing aperture. At sufficiently large apertures, the walls no longer influence the speed of the sphere and the tangential drag

experienced by the particle approaches that for a sphere settling in an unconfined fluid. For a water-filled fracture and 1 mm radius particle, a particle released in fractures with apertures >10 mm all exhibit nearly the same open-tank speed (Fig. 5b).

*Variable-aperture fracture*: Fractures in rock have heterogeneous aperture distributions[30–32]. Thus, chattering dust was released in variable-aperture fractures to explore how changes in the grain speed, determined by AE, is influenced by fracture aperture. Variable-aperture fractures where created by placing 2 mm thick rubber sheets with a designed aperture distribution between the same acrylic blocks, previously used for the uniform-fracture aperture. In Fig. 6, composite digital images (dimensions given in Supplementary Fig. 8c) are shown for each fracture and display the sequential locations of the dust grain in the fracture, as it fell under gravity. Four variable-aperture fractures were tested: a Y-shape, an inverted Y-shape, a diamond chain, and a converging aperture. The yellow circles in Fig. 6a indicate the sequential locations of a chattering dust grain as a function time for each of these fractures. For the Y-shape void aperture (Fig. 6b), the dust grain rests initially at the air–water interface, providing a stable reference point to calibrate the localization. Once it begins falling, it travels faster in the upper converging section (15.8 mm s⁻¹) of the fracture than in the lower 2 mm narrow channel (5.8 mm s⁻¹). For the inverted Y-shape-aperture fracture (Fig. 6c), the dust grain travels more slowly in the narrow 2 mm upper channel (3.2 mm s⁻¹) and then speeds up in the diverging aperture. As shown for particle swarms in diverging apertures[33], particles accelerate as fracture apertures diverge. The grain speeds in the variable-aperture fracture are slower than for the apertures shown in Fig. 5b because the in-plane aperture is 2 mm rather than the in-plane aperture of 150 mm aperture for the uniform-aperture case. The variable-aperture fractures result in an additional degree of confinement that affects the speed of the grain.

A key feature of the chattering dust is their ability to dissolve over time (see Supplementary Note 4). For instance, in the Y-shape channel (Fig. 6b), the dust grain was initially too large to fall into the 2 mm channel of the Y-shape aperture, and remained at the top of the neck for ~100 s. However, after dissolving to an appropriate size, the dust began its descent through the narrow channel and continued to emit (although reduced in size). This is observed several times for the diamond-chain fracture (Fig. 6d). This method could be developed as a calibration curve based on the rate of emission versus size to measure the size of apertures.

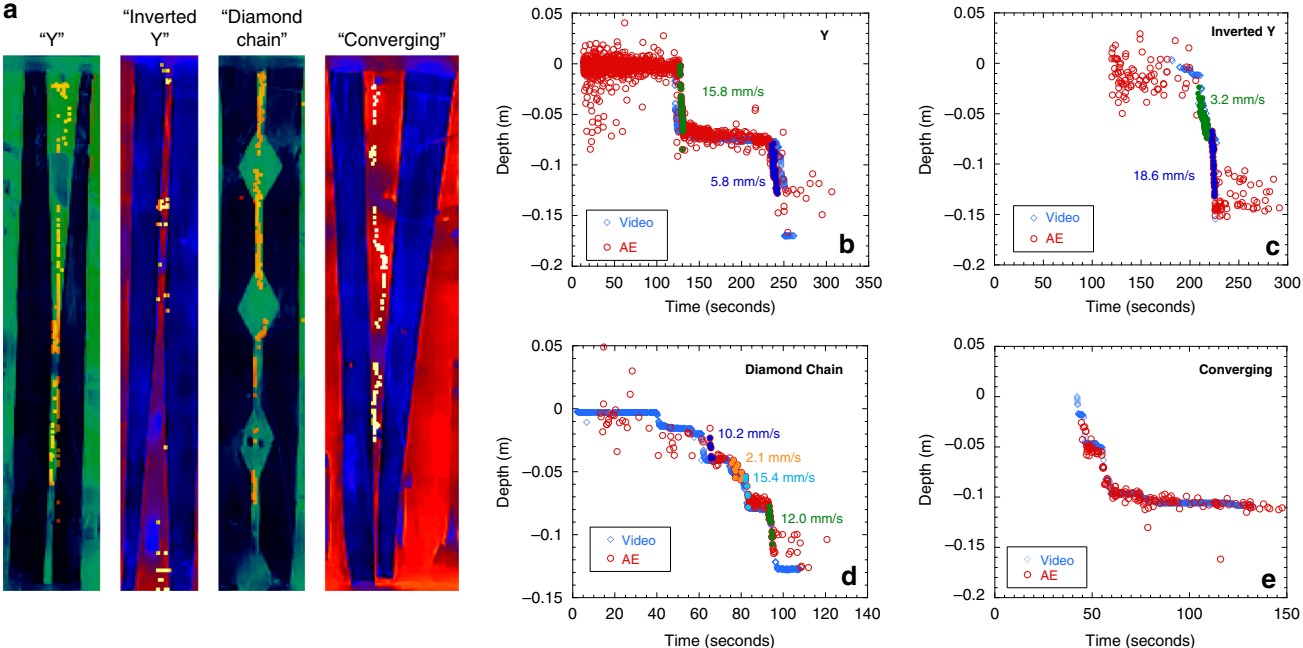

**Fig. 6 Dust speed controlled by fracture aperture. a** The false-color image of the actual fractures with yellow-orange dots indicating the location of the chattering dust during descent, (see Supplementary Fig. 8c for dimensions.) Location of chattering dust grains as a function of time from video imaging (blue diamonds) and acoustic emission (AE open red circles) are shown for **b** converging Y-shaped aperture, **c** diverging (inverted Y) aperture, **d** diamond chain, and **e** narrowing-aperture fracture (converging). Out-of-plane aperture was 2 mm in all cases.

As the chattering dust dissolves, the dust grain falls more slowly. Calibration of the rate of acoustic emissions as a function for particle size could help to fine-tune aperture estimates.

In the converging aperture (Fig. 6e), the speed of the dust grain varied with depth. From video images, the descent of the grain was interrupted between 47 and 55 s at which time the grain made contact with the wall. The path is shown in the composite image from the video of the converging channel with the path indicated by the yellow pixels. The overall depth-versus-time curve follows the functional form for a sphere falling between infinite parallel plates[28,29], i.e., a gradual deceleration of a sphere as the aperture narrows.

These experiments on uniform and variable-aperture fractures show two key features of chattering dust: (1) the descent speed is correlated with changes in the fracture aperture (Figs. 5b and 6), and (2) even if the dust is larger than a down-stream aperture, it eventually dissolves to a size that is able to fit into the aperture, while still emitting a signal (as demonstrated in Fig. 6a). Using the measured dissolution rates and the location of the particle as a function of time, aperture sizes can be estimated. The data from the uniform and variable-aperture fractures provide the first step for interpreting aperture sizes in a fracture having an unknown aperture distribution. These studies show that the chattering dust speed is a function of aperture and of the degree of confinement provided by the void geometry.

**Flow of dust through intersecting fractures**. An important application of chattering dust is the possibility to track dust grains, as they are transported through fracture intersections. By changing the fluid flow conditions, it is possible to "direct" the grains into one fracture or another. The direction of flow in intersecting fractures was controlled using a "T" intersection between two fractures (Fig. 7 and the Supplementary Note 7 for experimental details). A single chattering dust grain was released at the center of the fracture, as a flow field was established by creating a pressure difference across the horizontal fracture either

flowing from left to right (blue in Fig. 7a) or right to left (red in Fig. 7a). Tracking the acoustic emissions from the chattering dust easily followed the direction of the flow. In further experiments, the chattering dust was released at different $x$-locations (Fig. 7b, c) along the top of the fracture, as a flow field was established from right to left in the fracture through central ports on the inlet and outlet sides of the fracture. In all cases, the chattering dust was drawn to the center of the fracture (Fig. 7b, c) because flow lines focus into the central portion of the fracture. In another experiment (Fig. 7c–f), the dust was released from a positive $x$ position while the flow inlet also was located at $a + x$ position, but the outlet was at a $-x$ position (Fig. 7f). Therefore, the grain trajectories in Fig. 7c–e follow the local flow lines within the fracture (with an error in location of +1 cm), mapped out using the self-acoustic localization of the chattering dust grains.

## Discussion

As presented in this paper, chemically induced microseismicity combined with particle transport provides a unique opportunity to locate dominant flow paths at the laboratory scale by tracing the acoustic emissions within a laboratory fracture or fracture network. Chattering dust takes advantage of the well-developed field of acoustic emission and induced seismicity, hence using chattering dust for laboratory experiments on rock enables more complex systems to be studied than is possible using X-ray or optical imaging. We calibrated the speed as a function of the aperture size making it possible to estimate aperture sizes, as chattering dust grains move through variable-aperture fractures. For instance, in the variable-aperture fractures in Fig. 6, changes in aperture resulted in acceleration and deceleration of the emitters as they moved from small to large apertures (Fig. 6b) or from large to small apertures (Fig. 6a), respectively. Like other transportable emitters (e.g., ref. [34]), a chattering dust grain cannot enter a fracture aperture that is smaller in size than the dust (e.g., Fig. 6a). However, the transportable emitters in this study have the unique property that they slowly dissolve, which enables them

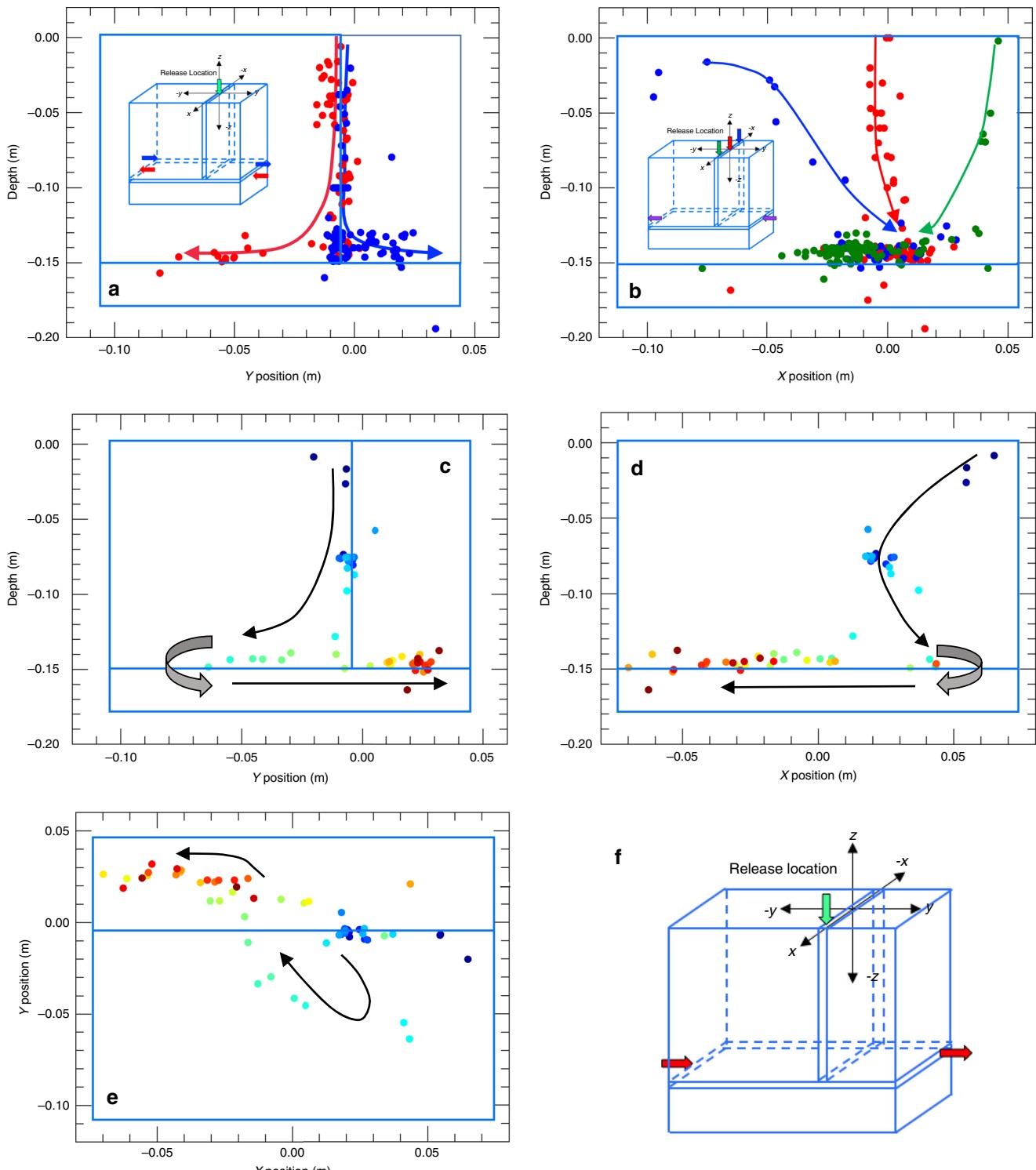

**Fig. 7 Tracking streamlines in intersecting fractures.** Flow through an inverted T fracture intersection for: **a** dust $z$-$y$ location as a function of horizontal location during flow to the right (blue) and to the left (red), **b** $y$-$x$ location for different release locations during flow to the left, and **c-e** complex flow paths as chattering dust is transported through the vertical fracture and into the horizontal fracture, which was subjected to flow from left to right (color of symbols represents early time (blue) to later time (red)). The release location of the dust for **c-e** is indicated by the green arrow in **f**, and the red arrows show the inlet and outlet locations for the fluid.

to reduce in size until they can be transported through a restriction. Furthermore, the tangential drag from the fracture walls depends on the ratio of the particle diameter to distance from the wall (Fig. 5). As a dust grain continues to dissolve, this time dependence could be used as a form of internal calibration.

Perhaps the most important capability of chattering dust is its ability to track the connected flow paths and flow field lines in fractures. For example, in the "T" intersection flow experiments, the dust grain followed the direction of flow through the horizontal fracture (Fig. 7a). Dust also enabled the identification of

flow field lines (Fig. 7b–e) within the vertical fracture and the horizontal fracture. Currently, there are no other methods that can determine the flow field lines in opaque materials, such as rock or the direction of flow through more complicated fracture networks in rock cores or other geotechnical laboratory samples.

Extending chattering dust applications into the field could occur in two settings: nondestructive testing of civil infrastructure and near-borehole geophysics. The existing chattering dust sources and receivers are already capable of transmitting through concrete structures up to 2 meters using 20 kHz receivers (see Supplementary Fig. 11 in Supplementary Note 9). The first field demonstrations on civilian infrastructure may help find the extent and depth of cracks in buildings, dams or parking structures, where it is easy to distribute the dust and to place sensors. For near-borehole applications, the use of DAS (distributed acoustic sensing) systems, that are sensitive to strains on the $\sim 10^{-8}$–$10^{-9}$ range[35], could detect chattering dust through several meters of rock. In the future, deployment in a broader range of geophysical applications may require improved dust lifetime through slower dissolution rates (involving new chemical formulations), development of ultralow energy sensors and discrimination software, the use of swarms of particles to enhance the signals, or improved dust matrix yield strength and higher gas pressures to release more energy. Currently, a single gas-inclusion emission in this study has the equivalent of 0.4 micro-Joules of energy ($E = PV$, where $P$ is pressure and $V$ is the volume of the bubble), which is equivalent to a moment magnitude $M_w = -10.0$ earthquake (see Supplementary Note 8 for calculation). More energetic sources could help identify 3D fracture flow paths to mitigate leakage paths from anthropogenic waste isolation sites, or to map fracture flow paths that affect the production and safety of water in fractured aquifers. Although such large-scale field applications will require further development, the transportable, active and chemically reactive dust available today is already poised for applications on the meter scale.

## Methods

**3D X-ray microscopy**. A 3D X-Ray Microscope (Zeiss Versa 510) was used to acquire 2D projections to perform 3D computed tomography on individual single dust grains to measure the bubble volume probability and spatial distributions, and to determine the number of bubbles in a single grain. The settings used to acquire the X-ray images are give in Supplementary Table 1 in Supplementary Note 1 along with the pixel resolution (or voxel edge length) for each sample.

**Acoustic emission**. AE sensors were connected to an AE measurement system (24 Channel Mistra Express or an 8 Channel Mistra Express) through preamplifiers (Mistra 1220-5054, 20/40/60 dB single-ended powered preamplifier) to record signals using Mistra AEWin software. Depending on the experiment (see Supplementary Notes 2–7 and 19), the threshold amplitude for detection was set to a value between 25 and 45 dB (with a 60 dB preamplifier setting with 100–400 Khz window), which was determined to eliminate ambient noise. An acoustic emission was recorded when the signal amplitude exceeded this threshold. AE locations were interpreted using either the Mistra AEWin software or custom codes (see Supplementary Note 6).

**Digital imaging**. An imaging system was used that consisted of a Spy camera (Spy camera for Raspberry PI No 1397 from Adafruit) connected to a Raspberry Pi Model B + with 512 MB RAM that captured three layered images (red, green, and blue (rgb) arrays. Image dimensiosn wee 2592 pixels × 1944 pixels acquired at a rate of 1 frames per second (f.p.s.) for the dissolution experiments (see Supplementary Note 4), at 2 f.p. s. for the seeded fracture experiments (see Supplementary Note 5) and at a rate of 29 f. p.s. (with dimensions of 280 pixels by 480 pixels) for the uniform- and variable-fracture aperture experiments (see Supplementary Note 6).

## Data availability

Data are available upon request from L.J.P.-N. at ljpn@purdue.edu.

## Code availability

Analysis codes are available upon request from ljpn@purdue.edu.

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

## Acknowledgements

The authors acknowledge support of this work by the U.S. Department of Energy, Office of Science, Office of Basic Energy Sciences, Geosciences Research Program under Award Number (DE-FG02-09ER16022). We also acknowledge support from NSF-REU grant (PHY-1460899 at Purdue) for summer salary support of W.B., support from the EVPRP Major Multi-User Equipment Program 2017 at Purdue University for acquisition of the Zeiss Versa 510 3D X-ray Microscope, Dr. Timothy J Kneafsey for the acrylic fracture replica, assistance from C.A. Mitchell and L. Jiang with X-ray image capture, segmentation and analysis, and W.H. Casey for measuring the acoustic properties of the transducer couplant.

## Author contributions

L.J.P.-N., D.D.N., and N.J.N. conceived the experiments; W.B., N.J.N., A.W., and L.J.P.-N. conducted the experiments; L.J.P.-N. and D.D.N. analyzed the results. All authors reviewed the manuscript.

## Competing interests

The authors declare no competing interests.
