## [Peer Review File · Nature Communications]

Reviewers' comments:

Reviewer #2 (Remarks to the Author):

Major Claims: Our energy and environmental systems rely on the flow of fluids through fractures. Roughly 85% of our energy still is from subsurface extraction of fossil fuels making fracture flow in subsurface systems a very impactful, broad topic of interest. However, fracture flow in the subsurface cannot be imaged directly since these operations generally take place thousands of feet below the ground making it difficult to optimize or control fractured systems. This manuscript employs an innovative approach to utilize "chattering dust" to track chemically-activated dust particles that emit acoustic emissions as they dissolve and flow through a fracture network. The key advantage of the approach is that it has the potential to delineate the transport path through a fracture network which would be transformational for optimizing and eventually controlling fracture flow in subsurface systems. It is well understood that fluids only access a small percentage of fractured rocks and we have very little knowledge of which fractures actually flow in subsurface systems. The approach combines ideas from tracer testing and remote geophysical monitoring to illuminate connected flow paths through a fracture system.

Conclusions: The paper demonstrates a compelling proof of concept of using chattering dust to determine flow paths through a fracture network in the laboratory. The authors state that a key question of the approach refers to scalability of the approach. I would like to see more discussion on scalability since it is key to the eventual utility of the approach in the field. Detecting chattering dust at the lab scale versus trying to detect acoustic emissions that are generated thousands of feet below the ground in noisy environments is a very different challenge. For example, oil and gas companies have been trying proppants that emit acoustic emissions to determine how far proppants transport into a fracture system to determine their effectiveness. So far these methods have been ineffective in the field.

It is true that we currently have no reliable methods to illuminate 3D flow paths even at the lab scale through opaque rock samples. I would like more discussion on why knowing 3D flow paths even at the lab scale pushes the field forward since moving to the field scale seems to be a big leap. I believe illuminating 3D flow paths at the lab scale is already an impactful result since apertures that can be measured at the lab scale (micron-mm apertures) can conduct enough flow to be relevant in unconventional oil and gas production during late production times or in leading to leakage from carbon sequestration sites that rely on low permeability caprocks to contain CO₂. If moving to the field is not a big leap, then this work is very impactful. I'd like more discussion on how this technique could eventually be deployed in the field. It is ok even if multiple steps need to be taken and proven to eventually get there. The authors do provide numbers on the magnitude of the emissions and what is detectable. Therefore, expanding this discussion would be of interest to readers.

Methods: In general, I found the method technically sound and well explained. As far as the technique, the chattering dust is always changing as it flows and dissolves through a fracture network. It seems like this would make interpreting the results of these experiments more challenging. Is the current interpretation technique of the acoustic emissions result in a unique delineation of the flow path through the system? Also due to the complex nature of the chattering dust, are the experiments reproducible? Some comments on reproducibility should be provided.

In summary, I found this to be an innovative paper on an impactful topic that I believe should be published in Nature Communications if my comments are addressed. I believe the paper would be greatly strengthened if more of the advantages and drawbacks of the methods at various scales were spelled out more clearly. If this technique is limited to the lab scale, is this enough of success?

Reviewer #3 (Remarks to the Author):

Please see the two attached PDF:

General comments (Major comments): [FINAL] Reviewer_Comments.pdf

Annotated PDF (major and minor comments): [FINAL] Annotated Comments.pdf

Reviewer #4 (Remarks to the Author):

Report on: "Probing Complex Geophysical Geometries with Chattering Dust" submitted to Nature Communication

General comments to Authors: The manuscript describes application of geophysical acoustic methods –acoustic emissions-to track chemically-activated "chattering" dust particles as they flow through fractures. Authors claim that with monitoring the acoustic emission's source locations they could infer aperture distribution of fractures or interfaces.

The core idea is to track the source location of "chattering" particles while they go through fracture. In the following I will argue the method might not be accurate and there are some flaws in the described method and employed techniques.

1-how dissolving reactive grains accurately interact with the fracture topography's complexity – considering that pressure carbon dioxide (4.1Mpa) and the coating quality which suppose to dissolve? How this is dependent on the speed of fluid flow and interaction/hydro- mechanical interaction of a grain or collective of grains with the fracture surface?

2- Another important point which I believe authors ignored is to discuss the accuracy of their source locations as well as characterization of amplifier-sensor response to see how signal is distorted or does have phase shift.

The acoustic emission sources could be very different ranging from structural phase transition to different types of defects. Here as I am sure most of the events are from explosion of grains but the time line of explosion which occurs in the most converging point of the fractures (?) would restrict the application of the method. In fact, a single grain might generate many events and in this point it is not clear to me how a single grain (precise chemical or mechanical formulation is needed for this) emits acoustic phonons; the rate of emissions as well as their characteristics have not been studied and it is not clear how one can distinguish these signal from other corrosion or flow induced signals

Apart of the above points (and many other points), I do not see any fundamental or intriguing point in this paper regarding both natural emissions or study of complexity of fracture topography (or aperture map).

Reviewer's comments to the Authors

In the manuscript entitled: "Probing Complex Geophysical Geometries with Chattering Dust" by Pyrak-Nolte et al., the authors attempt to deploy so-called "chattering dust" in five different experimental configurations. In these experiments this novel acoustic source is deployed in an attempt to probe properties of fluid transport along analog experimental faults. These types of studies are very important to both geophysical and geomechanical fields of studies. Some quantities they attempt to quantify is the rate of fluid intrusion into a dry, rough analog fracture, the particle velocities of particles driven by gravity in idealized fault with increasingly complex cross-sectional profiles, flow of fluids in matrix-fracture interfaces and complex fluid flow fields occurring at an "inverted-T" fracture joint.

Principle of the novel sensor: Chattering dust is a novel organic acoustic source composed of a sucrose-based body filled with many (1000s to 10,000s) spherical inclusions of pressurized CO₂ gas ($P_{pore} \sim 4.1$ MPa). The authors' showed that when the a Dust particle comes in contact with water, the sucrose body dissolves, likely converted to an aqueous solution, and, in certain cases, the pressurized pores suddenly discharge energy as the pore pressure equalizes to the ambient surrounding. This sudden discharge makes an explosive acoustic (fluid) source, or it "*chatters*", within the fluid. This will cause pressure waves to propagate (discussed later as a major comment) that travel through a fracture, interacts with the wall and results, then propagate through the solid material govern by the equations of motions. Between 8-15 passive acoustic emission sensors are used to locate the sources (chatter) using the first (body wave) arrivals and a basic triangulation approach common in the field of laboratory acoustic emission studies.

While the idea is novel and shows tremendous potential the level of detail needed to develop, calibrate and deploy the "chattering dust" sensors has not been achieved. While the authors do attempt to characterize the sensor through the work described in Figures 1 to 3, it is, in my opinion, insufficiently characterized to deploy in the following experiments:

1. fluid intrusion into a "seeded-fault" configuration after Su et al. (1999) (Figure 4),
2. gravity-driven motion of a bluff particle in a fracture of constant aperture (Figure 5),
3. gravity-driven motion of a bluff particle in a fracture of variable aperture (Figure 6),
4. matrix-fracture interaction (Figure 7),
5. fluid flow into an inverted T section (Figure 8).

A major concern is that – at every step – the research is poorly described and the behavior of the chattering dust is not well-understood. The amount of experimentation is impressive but none of these five test have been sufficiently documented (either in the text or in the "Methods" section) to allow for reproducibility of these results with a degree of certainty. When I originally agreed to review this manuscript, I assumed that the full behavior of the chattering dust particles would have already been well-characterized in another, more technical, journal publication. I feel that Nature Geoscience is not the correct avenue to fully develop a sensor, then deploy it in five different experimental configurations. While all the experiments attack important and long-standing question related to fluid flow through fractures, there is a lack of clarity and transparency that is especially and important when performing scientific experiments. This paper is not suited in its current state for publication in Nature Geoscience. My suggestions is that the authors present a more technical paper describing more clearly the calibration of chattering dust. They should focus on the more detailed stepwise explanation on how acoustic sources are processed. More elaborate and accurate depiction of the Dust behaviors of chattering dust before attempting to show the results in this venue. Even with that said, to this date I have not seen a publication

in Nature Geoscience that attempts to deploy so many different experimental configurations when the space given is limited. This paper appears to be an “overview” of many experiments but is actually the first attempts to deploy this chattering dust to characterize flow of fluids in faults.

I should state that I think that chattering Dust is highly novel and an ingenious approach to active source fluid tomography using acoustic emissions. However, I do not feel it is accurately understood or described in this manuscript. Below are some notes on the technical concerns I have.

Decision: Decline publication on the basis of major technical and/or interpretational problems

Major concerns:

Describing the properties of Chattering dust:

A. Methodology surrounding the X-Ray Computed Tomography:

The methodology surrounding the XRCT scanning is not clearly presented or sufficiently described to make this experiment repeatable by others. Please provide the setting, scanning apparatus, resolution of the scans. As it stands, this is not an acceptable description of the methods.

Please consult Pini and Madonna (2016). They discuss the different techniques required to use a medical X-ray CT (mCT) scanning instrument (General Electric hi-speed CT/i X-ray computed tomography) or the Synchrotron-based X-ray radiation (μ CT). As they note, the types of CT will vary and how the data is processed will be machine- and user-dependent. Please also consult minor comments regarding Figure 1 in the annotated PDF and summarized below.

EDIT: I later found that the authors have referenced the scanner model in the acknowledgments: "...Purdue University for acquisition of the Zeiss Xradia 510 3D X-ray Microscope". This still does not sufficiently describe the methods used to describe how the authors have measured the sizes of CO₂ pores within the sucrose grain. I have looked up the maximum resolution Zeiss Xradia 510 3D X-ray Microscope. According to the data sheet, the spatial resolution can range from 0.7 microns to 70 nanometer. This will undoubtedly impact the smallest visible bubble size (3 microns diameter) and therefore needs to be mentioned?

Figure 1:

- It is unclear if these dimensions due to the lack of axes in the Figure 1(b) and the scan does not appear to be 0.9 mm x 0.9mm from visual interpretation of the aspect ratio. Perhaps showing the full scan of the sample and a visual image of the dust will be more informative than what is shown currently in this image. I do not feel that the 2D projection is very illustrative, and if this is a portion of the scan that was used for the tomographic reconstruction, can the authors show in Figure 1(b) where this projection plane is?
- Is it a single grain? If so the caption reads: "gas bubbles in reactive **grains**". Please clarify this point. The dissemination between a single and multiple grains are important since in Figure 2 the authors are showing the average acoustic behavior of single grains. In Figure 3, it appears to

show how amalgamated grains breakdown in the presence of water.

- While this is not part of the main experiment, the "calibration" and understanding the "acoustic event to bubble size" relationship is crucial to interpret the experimental results later. I do not feel convinced that the authors have described this sufficiently for other to reproduce these results.
- Please place axes on both images. It will help the reader if the 3D edges to the scan is given. As mentioned above. The methodology surrounding the scan is not well described in the body of the text or the "methods" section.
- This large dominant inclusion does not appear in the Figure 1(b) which leads me to think that Figure 1(a) is not used in the 3D reconstruction. This is fine but it does illuminate the point that there is a wide range of distribution of CO₂ pores sizes.

In Figure 3(a) the authors choose only to use the "average bubble size". I think that a standard deviation on this metric should also be described in the graphic (see note on figure 3). Perhaps the authors should show the probability distribution function (PDF) as a part of Figure 1.

Moreover, the authors need to clarify what "bubble size" means. Visually, it appears that the bubble form spheres and they later use an approximate diameter to describe the yellow bubble. I believe that the authors should be able to describe the PDF. While this point may seem moot, I note that:

ENERGY = PRESSURE x VOLUME.

If we assume that all pore are spherical and contain CO₂ pressurized at ~4.1 MPa, then the energy released will scale as

ENERGY \propto DIAMETER³.

Where the energy will determine the intensity of the recorded seismicity (assuming constant seismic efficiency, see e.g. Aki and Richards, 2002). More discussion on the true source will be mention after the major comments on the Dust calibration methods.

B. Relating the measure bubble size distribution to acoustic hits

I am unconvinced this is the best manner to present the data. In the annotated PDF, I have shown how the probability distribution function (PDF) of the bubble sizes and counts can be shown using a similar layout. I feel that perhaps graphically showing both un-normalized and normalized PDFs might give the reader a better understanding of the bubble sizes which are important and might help understand the relationship between Figure 2(a) and (b). Based on my remarks above. It is important to determine the average hits versus bubble size to see if larger detections are attributed to larger average pores since there appears to be no clear trend -- and the authors do not comment on a trend -- between the hits and number of bubbles or average bubble size.

C. Understanding the relationship between source (hits) and the decay of agglomerated dust particles

Again, there is no detail given to the experimental methods made here making reproducibility impossible for other researchers. This is a key experiment detailing how the dust particle decay in volume (size) while emitting sources – a key point in the almost all the subsequent experiment (but less so experiment 1). From my original comment on Page 1, this is why I think a more technical publication is necessary.

This portion of the calibration methodology appears flawed in my opinion and, again, does not describe the methods appropriately. Firstly, the authors appear to place multiple dust particles – making an agglomerate – into the test chamber based on the size of the of the red area on the inset picture at taken at 10 s. The size of this amalgamate is much larger than the average dust particle (describe earlier as ~ 1mm). This has inherent implications on all tests where the particle is moving (tests 2 to 5). We are not shown how a single particle behaves here which is what is used to sample fault and fluid flow properties in those experiments.

Moreover, another lack of methodology details comes in the form of how to calculate the decay of the area fraction with time. The authors mention that from the initial amalgamate of Dust particles will “disaggregate” upon the introduction of water -- clearly seen between the inset images taken at 10 and 100s. Due to the mobility issues associated with the flow of particles within a fracture, how these particles penetrate the fault will be highly dependent on how they disaggregate or simply shrink as the particle dissolves. Omitting that these particles also have an important non-linear and time-dependent emission (or so-called “hit”) behavior, simply the evolution of the particle size versus time is not well-understood here. As they mentioned, the particle size will have huge implications in terms of the Laplace-Young equation describing a particle behavior between two parallel plates.

A closer look at the non-linear and time-dependent emission relationship shown as blue line in figure 3, shows that the amalgamates response empirically fits the following equation:

$$N = 150 \exp(-0.015*t),$$

where N = hits/second. Integrating this we see that approximately ~10,000 hits were recorded over this experiment. From their original claims that grains are ~ 1 mm in length scale and produce ~ 100s to 1000s events. Can the authors please clarify exactly how many grains where originally placed in the chamber? Will a similar decay be observed for a single particle?

Please clarify for Figure 3:

- (y-axis) Number of signals (/sec)?
- Are "signals" equivalent to AE hits from Figure 2?

Experiment 1: “Seeded” complex rough-rough fracture flow

The authors use an identical experimental configuration to that used by Su et al. (1999). This is very poorly described and due to the lack of explanation make the experiment entirely inappropriate. For example, the authors simply show the aperture for the synthetic faults generated by Su et al. (1999). Upon closer inspection of the Su et al (1999) paper, I found that there are far more parameters needed to properly

convey the experiment the authors are trying to described in this paragraph (inclination angle, flow rates, etc.).

- The authors' state: Fracture aperture controls how a fluid invades a fracture. I believe other factors will control fluid intrusion: Dynamic viscosity, fluid density, pressure gradients, flow rate and micro-mechanical contact forces also have an influence on the fluid invasion process.
- The authors give no indication to how they seed the fault. This is a key point.
- From Su et al. (1999) the authors have concluded that the mean aperture height is 160 micrometers +/-110 micrometers. Below is the PDF of the aperture from Su et al. (1999). The fracture aperture is ~ 5 to 27% the average size of a single grain of dust. Are the authors compressing and fracturing dust particles along the interface? Fig 3 below Su et al. (1999) shows probability distribution function for the acrylic analogs. It shows that virtually none of the aperture measurements are larger than the mean Dust particle (~1mm). This begs the question, how were the particles seeded?

Figure 3. Probability distribution functions (PDFs) of epoxy fracture replica aperture distribution with and without confining gas pressure. Arithmetic mean aperture without confining gas pressure is 0.16 mm and with confining gas pressure is 0.17 mm.

Results in Figure 4(a) are difficult to justify since the methodology they used to map the seismic front is not detailed enough. They have not shown concomitant experimental results for acoustic hits and locations and the fluid penetration front. Was it not the point that the fluid front was visually observed by Su et al (1999). This is not performed here so I cannot understand how the authors are claiming this.

Disseminating the fluid front: It is unclear as to how the front was determined. Based on the calibration curves shown in Figure 3, regions where water has penetrated initially will become a continuously emitting source quite frequently for the first 100 seconds. Again, the methods described do not seem to accurately account for the fact the dust will continue to emit at locations where the water has already penetrated. The authors note that the red circles are locations of the events but neglect to mention how they separate the newly activated Dust particles at the fluid front from the previously activated particles using the raw acoustic records. I do not believe the "seeded" fault experiment is sufficiently described by the authors. Perhaps this can be done through back projection, which utilizes the time-reversal property

of seismic waves to retrieve their sources (see Marty et al., 2019 and references therein) but nothing to this effect is mentioned or described.

Another major concern I have, is how can the authors confirm that as the dust particles decrease in size (disaggregate and dissolve) they do not become mobilized and travel at or behind the fluid-air interface?

Why did the authors cut the bottom of the aperture field in Figure 4(b) from that used by Su et al. (1999) this seems misleading.

Experiment 2: Gravity-driven motion of a bluff particle in a fracture of constant aperture

Dissolving or disaggregating particle: My major concern is that we have not been shown how a single particle behaves in the calibration section. While figure 2 does show some relationship between bubbles and emissions this is not as important as the single dust particle size evolution versus time (Figure 3 shows how an amalgamation of dust particles behaves). Obviously, in the context of the Laplace-Young theory presented, particle size and aperture are key parameters (also fluid viscosity, which likely changes over time or near the particles boundary layer as the sucrose changes from a solid to aqueous state but this is likely a second-order effect). Without this knowledge of the particles true behavior it is difficult to claim this has no effect.

Complex source: I have drawn a schematic representation of what general seeded fault might look like for experiment 1. The panel on the left shows that it appears that the intact dust particle must be crushed to fit in the aperture in the seeded fault experiment. In the right panel, if the fractured practice emits an explosive source and the grain is in contact with the wall, this might be different that if the particle has dissolved and no longer touches the solid fracture wall. Either way, the manner in which the acoustics are interpreted are treated very poorly in this study and I think a more rigorous calibration study is merited before conclusion are drawn.

Experiment 1

In contrast to the “seeded” particles in experiment 1, the other experiments (2,3,4 and 5) have their own wave propagation problems (detailed schematically below).

While the Broyden Method is referred to in the Methods as the manner in which they solve the linear set of equations to determine the location, they do not take into account that the source might be more complicated than they propose.

From their logic, the particle falls due to gravity and is suspended in a fluid. The chemical reaction with the dust emits a source (or hit) which is therefore a suspended "point source" in a fluid. The pressure waves will then move through the fluid and interact with the fracture wall. Wave will then be produced in the acrylic blocks but assuming that the waves are always traveling through the acrylic solid (as they did when assuming a fault plane) is incorrect. You can also see on the inset images of Figure 6, the yellow orange dots that depict the path of descent (from the video) is not planar. This complex fluid structure interaction encountered by the elastodynamic waves needs more through calibration in a separate study.

Below is a schematic depiction of the two problems that the authors have not appropriately calibrated or considered in this study.

Experiments 2, 3, 4 and 5

The level of detail provided into the acoustic data processing in Figure 9 and the type of signals observed are not well-described. I would suspect that a more thorough study that described in a more rigorous manner how acoustic data looks and more details to each experiment would make the sources of errors and problems I mentioned above more.

Please see the Annotated PDF for minor comments.

Response to Reviewers

Manuscript: “Probing Complex Geophysical Geometries with Chattering Dust”

Authors: Pyrak-Nolte et al.

Table of Contents

A. General Comments to All of the Reviewers.....	1
B. Comments from Reviewer 2 and Response.....	1
C. Comments from Reviewer 3 and Response.....	4
C.1 General Comments.....	4
C.2 Annotated PDF Comments.....	17
D. Comments from Reviewer 4 and Response.....	25

A. General Comments to All of the Reviewers

The authors thank all of the reviewers for their comments and thoughtful questions. We include the requested experimental details and other information in the revised manuscript and figures. We now also include a detailed Supplementary Methods document on the characterization of the dust, details of the experimental approaches, analysis methods and interpretation methods, with a section on error analysis in the speed of descent. The seeded fracture experiments were re-done to include digital imaging of the entire fracture plane during imbibition of water into the fracture plane. We hope that we have addressed the concerns of the reviewers.

Please note: All figures labeled with “S” (example Figure S5 or Section S4) refer to figures/tables/sections in the new Supplemental Methods document. Answers to questions are in *italics* while sections from the manuscript or Supplemental Methods are contained within “ ” (quotes), as are parts of reviewer’s comments.

B. Comments from Reviewer 2 and Response

Major Claims: Our energy and environmental systems rely on the flow of fluids through fractures. Roughly 85% of our energy still is from subsurface extraction of fossil fuels making fracture flow in subsurface systems a very impactful, broad topic of interest. However, fracture flow in the subsurface cannot be imaged directly since these operations generally take place thousands of feet below the ground making it difficult to optimize or control fractured systems. This manuscript employs an innovative approach to utilize “chattering dust” to track chemically-activated dust particles that emit acoustic emissions as they dissolve and flow through a fracture network. The key advantage of the approach is that it has the potential to delineate the transport path through a fracture network which would be transformational for optimizing and eventually controlling fracture flow in subsurface systems. It is well understood that fluids only access a small percentage of fractured rocks and we have very little knowledge of which fractures actually flow in subsurface systems. The approach combines ideas from tracer testing and remote geophysical monitoring to illuminate connected flow paths through a fracture system.

Conclusions: The paper demonstrates a compelling proof of concept of using chattering dust to determine flow paths through a fracture network in the laboratory. The authors state that a key question of the approach refers to scalability of the approach. I would like to see more discussion on scalability since it is key to the eventual utility of the approach in the field. Detecting chattering dust at the lab scale versus trying to detect acoustic emissions that are generated thousands of feet below the ground in noisy environments is a very different challenge. For example, oil and gas companies have been trying proppants that emit acoustic emissions to determine how far proppants transport into a fracture system to

determine their effectiveness. So far these methods have been ineffective in the field.

It is true that we currently have no reliable methods to illuminate 3D flow paths even at the lab scale through opaque rock samples. I would like more discussion on why knowing 3D flow paths even at the lab scale pushes the field forward since moving to the field scale seems to be a big leap. I believe illuminating 3D flow paths at the lab scale is already an impactful result since apertures that can be measured at the lab scale (micron-mm apertures) can conduct enough flow to be relevant in unconventional oil and gas production during late production times or in leading to leakage from carbon sequestration sites that rely on low permeability caprocks to contain CO₂.

If moving to the field is not a big leap, then this work is very impactful. I'd like more discussion on how this technique could eventually be deployed in the field. It is ok even if multiple steps need to be taken and proven to eventually get there. The authors do provide numbers on the magnitude of the emissions and what is detectable. Therefore, expanding this discussion would be of interest to readers.

Methods: In general, I found the method technically sound and well explained. As far as the technique, the chattering dust is always changing as it flows and dissolves through a fracture network. It seems like this would make interpreting the results of these experiments more challenging. Is the current interpretation technique of the acoustic emissions result in a unique delineation of the flow path through the system? Also due to the complex nature of the chattering dust, are the experiments reproducible? Some comments on reproducibility should be provided.

In summary, I found this to be an innovative paper on an impactful topic that I believe should be published in Nature Communications if my comments are addressed. I believe the paper would be greatly strengthened if more of the advantages and drawbacks of the methods at various scales were spelled out more clearly. If this technique is limited to the lab scale, is this enough of success?

Response to Reviewer 2's comments in red from the above paragraphs:

"I would like to see more discussion on scalability since it is key to the eventual utility of the approach in the field. Detecting chattering dust at the lab scale versus trying to detect acoustic emissions that are generated thousands of feet below the ground in noisy environments is a very different challenge."

R2.1 *The discussion now includes:*

"A key question is the translation of the technique from the laboratory-scale to the field. Deployment to a borehole or rock mass will require additional research to improve the life time of the dust (which may require new chemical formulations), the use of DAS (distributed acoustic sensing) systems that are sensitive to strains on the $\sim 10^{-8}$ to 10^{-9} range, new development of ultra-low energy sensors and discrimination software, the use of swarms of particles to enhance the signals, or re-engineering of the particles to release more energy. For instance, a single dust grain emission in this study has the equivalent of 1 micro-Joules of energy, which is equivalent to a magnitude -10.0 earthquake. A cloud of emitters could be used to enhance the signal for possible near well-bore integrity studies using borehole sensors. The most likely field demonstration would be related to civilian infrastructure such as finding the extent and depth of cracks in buildings, dams or parking structures because of ease of access and placement of sensors. On the laboratory scale, chattering dust takes advantage of the well-developed field of acoustic emission and induced seismicity, and using chattering dust for laboratory experiments on rock will enable more complex systems to be studied under stress than is possible using X-ray or optical imaging."

"I would like more discussion on why knowing 3D flow paths even at the lab scale pushes the field forward since moving to the field scale seems to be a big leap."

R2.2 *The discussion now includes:*

"Identifying 3D fracture flow paths could help to mitigate leakage paths from anthropogenic waste isolation sites, to identify cracks in civilian infrastructure such as dams or buildings or parking ramps, to know the extent of induced fractures in rock (to see if potential penetration of existing aquifers is possible), and to map fracture

flow paths that affect the production and safety of water in fractured aquifers.”

“Chattering dust can also be used to illuminate the connected flow paths and flow field lines in fractures. For example, in the “T” intersection flow experiments, the chattering dust followed the direction of flow through the horizontal fracture (Figure 7a). Dust also enabled the identification of flow field lines (Figure 7b-e) within the vertical fracture and the horizontal fracture. Currently, there are no other methods for determining the flow field lines in opaque materials such as rock or the direction of flow through more complicated fracture networks. “

“I’d like more discussion on how this technique could eventually be deployed in the field. It is ok even if multiple steps need to be taken and proven to eventually get there. The authors do provide numbers on the magnitude of the emissions and what is detectable. Therefore, expanding this discussion would be of interest to readers.”

R2.3 *See the response in R2.1 which includes a discussion of what is needed for the field.*

Methods: In general, I found the method technically sound and well explained. As far as the technique, the chattering dust is always changing as it flows and dissolves through a fracture network. It seems like this would make interpreting the results of these experiments more challenging. Is the current interpretation technique of the acoustic emissions result in a unique delineation of the flow path through the system? Also due to the complex nature of the chattering dust, are the experiments reproducible? Some comments on reproducibility should be provided.

R2.4 *To address the question of repeatability we have included a Supplemental Method document that contains additional Figures and Tables with data for the reader to assess the repeatability of the experiments. Repeatability of the experiments is affected by dust grain volume, strength of interpretation method, as well as disaggregation of a particle. A section on errors attributed to the location method is now included in the Supplemental Methods (Section 6.4). For our AE system, errors in location are not a problem for apertures less than 2 mm with the error in location increasing to almost 3 mm for an aperture of 10 mm. However, the error in location translates to an error in the speed of descent, $V_{descent}$, of only 3%. Figure S9d addresses the effect of the volume of the particle on $V_{descent}$, which shows a linear increase in velocity with volume. With knowledge of the initial grain size and dissolution rates, estimates of apertures can be constrained. Obviously, different location methods can be tested to minimize or reduce location errors but this is beyond the scope of this manuscript whose purpose is to demonstrate the concept and benefits of chattering dust and potentially other future transportable sources.*

Is it enough for just laboratory work? For working on opaque systems where X-ray imaging is not possible, this provides a new method for tracking flow paths in fractured media as well as porous fractured media or in other devices such as complicated 3D microfluidic systems. (need more here).

R2.5 *While the method presented in this manuscript is not ready for field scale usage, this method is applicable to laboratory-scale studies (up to 1 meter) as now mentioned in the discussion for potential use in Civilian infrastructure. Currently, one can take a fracture in rock and measure the flow, but the details of the flow paths or connectivity of the voids in the plane is often unknown or inaccessible without X-ray imaging, especially under experimental conditions involving stress or fluid pressure. Therefore, chattering dust provides a non-destructive method for tracking flow paths and it can be performed repeatably as a fractured rock sample is subjected to stress or to variations in fluid pressures. Depending on the flow rates, chattering dust can be used repeatedly to build up strong statistics on the flow path geometry, and then the geometry can be changed through the application of stress and other*

physical and chemical processes. The altered geometry can be re-tested with chattering dust to examine the effect of alterations on flow. In addition, this work is exploring what it means to have a moving dissolvable source in a fracture network to provide a knowledge basis for future developments for the field. For example, as pointed out by reviewers 3 & 4, and addressed now in the supplemental methods section, the effect of refraction at the water-matrix interface on the interpretation of location is an interesting issue. Future work will examine how the presence of other fractures not connected to the fracture flow path affects interpretation. Chattering dust will provide a unique benefit for laboratory rock mechanics & microfluidics, and will provide the foundational science on interpretation of fracture location, extent, flow paths and connectivity for eventual field and infrastructure applications.

C. Comments from Reviewer 3 and Response

C. 1 General Comments

Reviewer's comments to the Authors

In the manuscript entitled: "Probing Complex Geophysical Geometries with Chattering Dust" by Pyrak-Nolte et al., the authors attempt to deploy so-called "chattering dust" in five different experimental configurations. In these experiments this novel acoustic source is deployed in an attempt to probe properties of fluid transport along analog experimental faults. These types of studies are very important to both geophysical and geomechanical fields of studies. Some quantities they attempt to quantify is the rate of fluid intrusion into a dry, rough analog fracture, the particle velocities of particles driven by gravity in idealized fault with increasingly complex cross-sectional profiles, flow of fluids in matrix-fracture interfaces and complex fluid flow fields occurring at an "inverted-T" fracture joint.

Principle of the novel sensor: Chattering dust is a novel organic acoustic source composed of a sucrose-based body filled with many (1000s to 10,000s) spherical inclusions of pressurized CO₂ gas (P_{pore} ~ 4.1 MPa). The authors' showed that when the a Dust particle comes in contact with water, the sucrose body dissolves, likely converted to an aqueous solution, and, in certain cases, the pressurized pores suddenly discharge energy as the pore pressure equalizes to the ambient surrounding. This sudden discharge makes an explosive acoustic (fluid) source, or it "chatters", within the fluid. This will cause pressure waves to propagate (discussed later as a major comment) that travel though a fracture, interacts with the wall and results, then propagate through the solid material govern by the equations of motions. Between 8-15 passive acoustic emission sensors are used to locate the sources (chatter) using the first (body wave) arrivals and a basic triangulation approach common in the field of laboratory acoustic emission studies.

While the idea is novel and shows tremendous potential the level of detail needed to develop, calibrate and deploy the "chattering dust" sensors has not been achieved. While the authors do attempt to characterize the sensor through the work described in Figures 1 to 3, it is, in my opinion, insufficiently characterized to deploy in the following experiments:

1. fluid intrusion into a "seeded-fault" configuration after Su et al. (1999) (Figure 4),
2. gravity-driven motion of a bluff particle in a fracture of constant aperture (Figure 5),
3. gravity-driven motion of a bluff particle in a fracture of variable aperture (Figure 6),
4. matrix-fracture interaction (Figure 7),
5. fluid flow into an inverted T section (Figure 8).

A major concern is that – at every step – the research is poorly described and the behavior of the chattering dust is not well-understood. The amount of experimentation is impressive but none of these five test have been sufficiently documented (either in the text or in the "Methods" section) to allow for reproducibility of these results with a degree of certainty. When I originally agreed to review this manuscript, I assumed that the full behavior of the chattering dust particles would have already been well-characterized in another, more technical, journal publication. I feel that Nature Geoscience is not the correct avenue to fully develop a sensor, then deploy it in five different experimental configurations. While all the experiments attack important and long-standing question related to fluid flow through fractures, there is a lack of clarity and transparency that is especially and important when

performingscientific experiments. This paper is not suited in its current state for publication in Nature Geoscience. My suggestions is that the authors present a more technical paper describing more clearly the calibration of chattering dust. They should focus on the more detailed stepwise explanation on how acoustic sources are processed. More elaborate and accurate depiction of the Dust behaviors of chattering dust before attempting to show the results in this venue. Even with that said, to this date I have not seen a publication in Nature Geoscience that attempts to deploy so many different experimental configurations when the space given is limited. This paper appears to be an “overview” of many experiments but is actually the first attempts to deploy this chattering dust to characterize flow of fluids in faults.

I should state that I think that chattering Dust is highly novel and an ingenious approach to active source fluid tomography using acoustic emissions. However, I do not feel it is accurately understood or described in this manuscript. Below are some notes on the technical concerns I have.

Decision: Decline publication on the basis of major technical and/or interpretational problems

The authors thank reviewer 3 for the detailed comments and thoughtful questions. We now include the information in the revised manuscript and agree that this is important information. In addition to revisions in the manuscript and figures, we have now also included a detailed Supplementary Methods document to address reviewer 3's major concerns on the characterization of the dust, details of the experimental approaches and analysis/interpretation methods. Our response to reviewer 3's comments are given below each comment.

Major concerns:

Describing the properties of Chattering dust:

1- A. Methodology surrounding the X-Ray Computed Tomography:

The methodology surrounding the XRCT scanning is not clearly presented or sufficiently described to make this experiment repeatable by others. Please provide the setting, scanning apparatus, resolution of the scans. As it stands, this is not an acceptable description of the methods.

Please consult Pini and Madonna (2016). They discuss the different techniques required to use a medical X-ray CT (mCT) scanning instrument (General Electric hi-speed CT/i X-ray computed tomography) or the Synchrotron-based X-ray radiation (μ CT). As they note, the types of CT will vary and how the data is processed will be machine- and user-dependent. Please also consult minor comments regarding Figure 1 in the annotated PDF and summarized below.

EDIT: I later found that the authors have referenced the scanner model in the acknowledgments: "...Purdue University for acquisition of the Zeiss Xradia 510 3D X-ray Microscope". This still does not sufficiently describe the methods used to describe how the authors have measured the sizes of CO₂ pores within the sucrose grain. I have looked up the maximum resolution Zeiss Xradia 510 3D X-ray Microscope. According to the data sheet, the spatial resolution can range from 0.7 microns to 70 nanometer. This will undoubtedly impact the smallest visible bubble size (3 microns diameter) and therefore needs to be mentioned?

R3.1 *Following the reviewer's request, section S1 in the Supplementary Methods (entitled: S1. Methodology for 3D X-ray Microscopy of Chattering Dust to Determine Bubble Distribution) contains information on the 3D X-ray microscope and the settings used for each dust particle scanned (Section S1.1. Table S1 includes energy, power, source & detector distance, optical magnification, exposure time, bin size and pixel size).*

Yes, the spatial resolution affects the interpretation of bubble size. For the 20x objective, the voxel edge length (or pixel edge length) was 0.9072 μ m. For the 4x objective the voxel edge length ranged from 4.2003 – 5.3894 μ m for the different single grain samples. There is a range of resolutions at 4x because the source/detector distances were adjusted to capture the entire grain and grain size varied (see bulk volume in Table S2). For interpretation of bubble volume, only bubble volumes with volumes \geq

$4\pi/3[5*\text{pixel resolution}]$ (assuming a sphere) where used in the analysis (i.e. at least 10 pixels in diameter). This of course limits the low end of the bubble volume distribution. All of this is now described in the Supplementary Methods.

Figure 1:

- It is unclear if these dimensions due to the lack of axes in the Figure 1(b) and the scan does not appear to be 0.9 mm x 0.9mm from visual interpretation of the aspect ratio. Perhaps showing the full scan of the sample and a visual image of the dust will be more informative than what is shown currently in this image. I do not feel that the 2D projection is very illustrative, and if this is a portion of the scan that was used for the tomographic reconstruction, can the authors show in Figure 1(b) where this projection plane is?

R3.2 Figure 1 has been re-drafted (see new figure below) and now only includes images from Sample G. Axes, color scale and grid size have been added to the 3D reconstruction. The figure caption provides the dimension of the 3D reconstruction. The perimeter of the several bubbles have been highlighted in the 2D projection and in the corresponding bubble in the 3D reconstruction. We do not have a “visual image” of the dust grain. The 2D projection remains in the figure as it provides an additional view or sense of the bubble density which is hard to capture in 3D. A probability distribution of bubble volume has been added for this sample.

Figure 2 in the manuscript and Figure S1 in the Supplemental Methods contain the bubble volume probability distribution, a 3D visualization of the center of mass of the bubbles for each entire sample, and the log energy probability distribution for 4 of the 9 samples scanned with the 4x objective. The log energy probability distribution is based on the amplitude measured by the Acoustic Emission (AE) system which is described in Section S2. Methodology for Relating Acoustic Hits to Bubble Size Distributions.

Revised Figure 1. (a) 2D projection (dashed yellow outlines reconstructed region in (b)). (b) 3D reconstruction of a subregion ($622.34 \times 797.43 \mu\text{m} \times 600.57 \mu\text{m}^3$) of a single grain taken with a 20x objective on a 3D X-ray microscope showing bubbles with radii great than 5 pixels. The color scale in (b) represents bubble volume. The red bubble in (b) is the bubble circled in red in (a). The diameter of the red circle in (a) is $243 \mu\text{m}$. The pixel resolution is given in Table S1 in Supplemental Methods. (c) Probability distribution of the log of bubble volume for bubbles shown in (b).

- Is it a single grain? If so the caption reads: "gas bubbles in reactive grains". Please clarify this point. The dissemination between a single and multiple grains are import since in Figure 2 the authors are showing the average acoustic behavior of single grains. In Figure 3, it appears to show how amalgamated

grains breakdown in the presence of water.

R3.3a *We have corrected the caption for Figure 1 (as well in other figures, descriptions and tables) has been corrected to indicate that it was a single grain.*

R3.3b *We have revised Figure 2 (see below) to show two graphs (a & d) of the histogram of bubble volume for two individual grains and companion graphs of the log energy pdf (b&e) with an inset of the bubble locations (c&f). Supplemental Figure S1 shows examples for 2 additional single grains. In addition, in Section 1.2 in the Supplemental Methods, Figure S2(left) shows the number of bubbles per bulk volume. Figure S2(right) shows the bulk volume and bubble count for each individual grain with the values listed in Table S2. These are the same single grains that were X-rayed as described in section S1.*

Revised Figure 2. (a&d) Probability distribution of bubble volume (in Log_{10}) for single-grain samples CD2A and CD6A (see Table S1 in Supplemental Methods). (b&e) Distribution of the logarithm of energy generated by the bubbles in the grain (number of recorded events shown in red and based on recorded events from one of the sensors). (c&f) 3D visualization of the center of mass of the bubbles within each single grain (number in red is number of bubbles) from X-ray imaging

R3.3c *We have added new information on the dissolution experiments in Section S4 in the Supplementary Method (S4. Methodology for Relating Dissolution Rate and Acoustic Events Events). Of the 12 grains tested, only 3 grains used in the dissolution experiments disaggregated. The dissolution behavior for grains that disaggregated differed. All grains were initially a single grain. Disaggregation depends on the thickness of sucrose between bubbles, the size of the bubble in terms of energy of release, and dissolution rates. (More on dissolution is given later in response to review R3.10, and in Figure 3 in the manuscript, and Section S4. Methodology for Relating Dissolution Rate and Acoustic Events in the Supplemental Method document.)*

- While this is not part of the main experiment, the "calibration" and understanding the "acoustic event to bubble size" relationship is crucial to interpret the experimental results later. I do not feel convinced that the authors have described this sufficiently for other to reproduce these results.

R3.4 *The new Supplemental methods gives a detailed description of the approach and analysis of the bubble volume to acoustic emission relationship (S2. Methodology for Relating Acoustic Hits to Bubble*

Size Distributions). In general, more acoustic emissions occurred in samples with more bubbles. The relationship is not 1:1 as shown in the Figure S4 (right) with more bubbles observed than AE hits for all AE channels. The deviation from a 1:1 line is attributed to the threshold used during the AE acquisition.

- Please place axes on both images. It will help the reader if the 3D edges to the scan is given. As mentioned above. The methodology surrounding the scan is not well described in the body of the text or the "methods" section.

R3.5 *Axes have been placed on the images (see revised Figure 1 above or in the manuscript). A new Supplementary Methods document has been added that includes the details of the X-ray scanning (S1.1 3D X-ray Microscopy Set-up) and the other sections (S1 –S2) on the experimental approach and data analysis methods.*

- This large dominant inclusion does not appear in the Figure 1(b) which leads me to think that Figure 1(a) is not used in the 3D reconstruction. This is fine but it does illuminate the point that there is a wide range of distribution of CO2 pores sizes.

R3.6 *This was a good catch by the reviewer as the 2D projection and the 3D reconstruction were from different samples. Figure 1 (a&b) has been revised to contain the 2D projection and 3D reconstruction from the same sample. Several bubbles are now highlighted for comparison and the reconstructed region is highlighted with yellow dashed lines.*

In Figure 3(a) the authors choose only to use the "average bubble size". I think that a standard deviation on this metric should also be described in the graphic (see note on figure 3). Perhaps the authors should show the probability distribution function (PDF) as a part of Figure 1.

R3.7 *We agree and have added pdf graphs to Figure 1c for the high resolution 20x objective imaging and in Figure 2 for the 4x objective results on single grains with additional graphs for 2 other samples (Figure S1) in the Supplementary Methods.*

Moreover, the authors need to clarify what "bubble size" means. Visually, it appears that the bubble form spheres and they later use an approximate diameter to describe the yellow bubble. I believe that the authors should be able to describe the PDF. While this point may seem moot, I note that:

$$\text{ENERGY} = \text{PRESSURE} \times \text{VOLUME}.$$

If we assume that all pore are spherical and contain CO2 pressurized at ~4.1 MPa, then the energy released will scale as

$$\text{ENERGY} \propto \text{DIAMETER}^3.$$

where the energy will determine the intensity of the recorded seismicity (assuming constant seismic efficiency, see e.g. Aki and Richards, 2002). More discussion on the true source will be mention after the major comments on the Dust calibration methods.

R3.8 *From the X-ray analysis, "bubble size" is now defined by the bubble volume with only bubbles with volumes $\geq 4\pi/3[5*\text{pixel resolution}]$ included in the analysis/discussions. This truncates the lower end of the pdf for the data acquired with 4x objective (Figure S1a&d). As such, it does not include bubble volumes observed in the data acquired with 20x objective (Figure 1).*

Figures 2b&e and S1b&e contain the probability distribution of the log energy based on the recorded signals from AE measurements while Figures 2a&d and S1a&d contain the pdf for the \log_{10} of the volume. These graphs now provide a comparison between the bubble size and the energy of release distributions. The acoustic threshold is not related to the threshold of the volume distribution. The AE is sampling smaller events than were captured in the histogram of bubble volumes. Overall, the distributions are similar.

B. Relating the measure bubble size distribution to acoustic hits

I am unconvinced this is the best manner to present the data. In the annotated PDF, I have shown how the probability distribution function (PDF) of the bubble sizes and counts can be shown using a similar layout. I feel that perhaps graphically showing both un-normalized and normalized PDFs might give the reader a better understanding of the bubble sizes which are important and might help understand the relationship between Figure 2(a) and (b). Based on my remarks above. It is important to determine the average hits versus bubble size to see if larger detections are attributed to larger average pores since there appears to be no clear trend -- and the authors do not comment on a trend -- between the hits and number of bubbles or average bubble size.

R3.9 *We agree. Figures 1 & 2 in the manuscript and Figure S1 in the Supplemental Methods now include probability distributions for both the bubble volume (size) and the log energy (taken as proportional to the amplitude) for single grains. Grains that contained a higher fraction of large volume bubbles roughly match with signals with more energy (i.e. larger amplitude).*

C. Understanding the relationship between source (hits) and the decay of agglomerated dust particles

Again, there is no detail given to the experimental methods made here making reproducibility impossible for other researchers. This is a key experiment detailing how the dust particle decay in volume (size) while emitting sources – a key point in the almost all the subsequent experiment (but less so experiment 1). From my original comment on Page 1, this is why I think a more technical publication is necessary.

This portion of the calibration methodology appears flawed in my opinion and, again, does not describe the methods appropriately. Firstly, the authors appear to place multiple dust particles – making an agglomerate – into the test chamber based on the size of the of the red area on the inset picture at taken at 10 s. The size of this amalgamate is much larger than the average dust particle (describe earlier as ~ 1mm). This has inherent implications on all tests where the particle is moving (tests 2 to 5). We are not shown how a single particle behaves here which is what is used to sample fault and fluid flow properties in those experiments.

Moreover, another lack of methodology details comes in the form of how to calculate the decay of the area fraction with time. The authors mention that from the initial amalgamate of Dust particles will “disaggregate” upon the introduction of water -- clearly seen between the inset images taken at 10 and 100s. Due to the mobility issues associated with the flow of particles within a fracture, how these particles penetrate the fault will be highly dependent on how they disaggregate or simply shrink as the particle dissolves. Omitting that these particles also have an important non-linear and time-dependent emission (or so-called “hit”) behavior, simply the evolution of the particle size versus time is not well-understood here. As they mentioned, the particle size will have huge implications in terms of the Laplace-Young equation describing a particle behavior between two parallel plates

A closer look at the non-linear and time-dependent emission relationship shown as blue line in figure 3, shows that the amalgamates response empirically fits the following equation:

$$N = 150 \exp(-0.015*t),$$

where N = hits/second. Integrating this we see that approximately ~10,000 hits were recorded over this experiment. From their original claims that grains are ~ 1 mm in length scale and produce ~ 100s to 1000s events. Can the authors please clarify exactly how many grains were originally placed in the chamber? Will a similar decay be observed for a single particle?

Please clarify for Figure 3:

- (y-axis) Number of signals (/sec)?
- Are "signals" equivalent to AE hits from Figure 2?

R3.10 *In response to this comment, the new Supplemental Methods includes a section (S4. Methodology for Relating Dissolution Rate and Acoustic Events) with the details of the dissolution experiments and analysis. To address the reviewer's concerns we added the following to section S4 on disaggregated particles versus signal grain:*

“After initiating the AE monitoring and image capture systems, a single dust grain was placed in the water-filled petri dish with tweezers. The water was changed between tests. Of the 12 samples tested, only 3 samples (samples DT13, DT14, DT15 in Table S5) disaggregated into 2 or more subparticles over time (e.g. DT14 in Figure 3d in manuscript). The functional form of the time rate of change of the cross-sectional area dA/dt , of the grains differed between particles that exhibited disaggregation and those that did not. Figure 3 in the manuscript shows representative grain area versus time curves for samples DT04 and DT14 along with the number of AE events as a function of time. Note that the number of AE events was binned into 1 second intervals. For DT04 (Figure 3a in manuscript), the image analysis could not resolve the grain for times greater than 146 seconds though acoustic emissions were still occurring. The jump in the curve for DT04 around 50 seconds occurred when the grain rotated and exhibited a different cross-sectional area.

When a dust grain did not disaggregate (DT04-DT12 Figure 3d in manuscript), the area and number of hits changed linearly in time. For samples DT04-DT12, the average dA/dt of a grain was $-0.115 \text{ mm}^2/\text{s}$ with a standard error of $0.014 \text{ mm}^2/\text{s}$. For particles that disaggregated (DT13-DT15), the area and number of hits decreased exponentially with an average decay constant of $-0.036 \text{ mm}^2/\text{s}$. Single dust grains with a large initial area exhibited a longer duration of acoustic emissions (Figure S6 left). The correlation between change in dust area and number of AE hits with time was 0.806 (Figure S6 right).”

In the manuscript, we have revised Figure 3 (see below) to include the behavior for particles that disaggregate and those that don't. We have also included graphs of the average signal amplitude (Figure 3c&f) where the amplitude of an AE signal is defined as the maximum positive or negative signal excursion during AE hit (this definition is given in Section 2 in the Supplemental Methods). This shows that number of hits is decaying but the average event amplitudes are roughly constant.

Revised Figure 3. (a & d) A digital image showing a dust grain sitting in a petri dish over an AE sensor, and segmented images of the dust for sample DT4 which did not disaggregate and sample DT14 which disaggregated. (b & e) Change in grain size area and number of hits per second as a function of time for samples DT4 and DT14. (c&f) Average signal amplitude as a function time for samples DT4 and DT14.

Experiment 1: “Seeded” complex rough-rough fracture flow

The authors use an identical experimental configuration to that used by Su et al. (1999). This is very poorly described and due to the lack of explanation make the experiment entirely inappropriate. For example, the authors simply show the aperture for the synthetic faults generated by Su et al. (1999). Upon closer inspection of the Su et al (1999) paper, I found that there are far more parameters needed to properly convey the experiment the authors are trying to described in this paragraph (inclination angle, flow rates, etc.).

R3.11 *We referred the reader to the paper of Su et al. (1999) for the characterization of the aperture distribution. The goal was not to repeat their experiments. The experimental conditions are different but provide a reference for the aperture distribution shown in Figure 4 in the manuscript (which now includes the full fracture plane). We now provide the details of our experiment in Supplemental Methods Section S5.*

- The authors’ state: Fracture aperture controls how a fluid invades a fracture. I believe other factors will control fluid intrusion: Dynamic viscosity, fluid density, pressure gradients, flow rate and micro-mechanical contact forces also have an influence on the fluid invasion process.

R3.12 *Yes, the reviewer is correct that other parameters affect fluid invasion. We have re-written the sentence to read:*

“Because fracture aperture is one of the controls on how a fluid invades a fracture, we expect to observe a non-uniform invading front when immiscible fluid displacement^{21–23} occurs in a fracture or pore network.”

- The authors give no indication to how they seed the fault. This is a key point.

R3.13 *Figure 4 in the manuscript now shows the location of the dust on the fracture plane. In addition, Section S5.1 contains a description of the seeding process:*

“The 8 sensors were attached to the top half of the sample. The lower half was seeded with single grains of chattering dust. The upper half was then placed on the lower half, and a gentle pressure was applied to ensure the two fracture surfaces were registered. This process led to crushing of the dust resulting in several dust fragments in some locations (locations with multiple stars in Figure 4 of the manuscript). Images of the initial seed location were taken on for single surface and after registering with the lower half of the sample for comparison.”

- From Su et al. (1999) the authors have concluded that the mean aperture height is 160 micrometers +/-110 micrometers. Below is the PDF of the aperture from Su et al. (1999). The fracture aperture is ~ 5 to 27% the average size of a single grain of dust. Are the authors compressing and fracturing dust particles along the interface? Fig 3 below Su et al. (1999) shows probability distribution function for the acrylic analogs. It shows that virtually none of the aperture measurements are larger than the mean Dust particle (~1mm). This begs the question, how were the particles seeded?

R3.14 *Yes, the methods now described in the Section S5.1, grains are crushed during assembly. (see response to previous comment R3.13.)*

Figure 3. Probability distribution functions (PDFs) of epoxy fracture replica aperture distribution with and without confining gas pressure. Arithmetic mean aperture without confining gas pressure is 0.16 mm and with confining gas pressure is 0.17 mm.

Results in Figure 4(a) are difficult to justify since the methodology they used to map the seismic front is not detailed enough. They have not shown concomitant experimental results for acoustic hits and locations and the fluid penetration front. Was it not the point that the fluid front was visually observed by Su et al (1999). This is not performed here so I cannot understand how the authors are claiming this.

Disseminating the fluid front: It is unclear as to how the front was determined. Based on the calibration curves shown in Figure 3, regions where water has penetrated initially will become a continuously emitting source quite frequently for the first 100 seconds. Again, the methods described do not seem to accurately account for the fact the dust will continue to emit at locations where the water has already penetrated. The authors note that the red circles are locations of the events but neglect to mention how they separate the newly activated Dust particles at the fluid front from the previously activated particles using the raw acoustic records. I do not believe the "seeded" fault experiment is sufficiently described by the authors. Perhaps this can be done through back projection, which utilizes the time-reversal property of seismic waves to retrieve their sources (see Marty et al., 2019 and references therein)

but nothing to this effect is mentioned or described.

R3.15 *To address the reviewer's concerns, new experiments were performed to simultaneously capture the fluid front location with imaging (0.5 fps) to compare with AE events. This is described in Section S5 in the Supplemental Methods. We have also updated Figure 4 in the manuscript to include the aperture distribution from Su et al. for the full fracture plane, a digital image showing the location of the dust, and a composite image that shows the location of the fluid front as a function of time and the AE events as a function of time.*

R3.16 *Yes, the grains continue to emit over a duration of 1-2 minutes and the duration of the invasion process was on the order of 25 minutes. The data are now shown as a function of event time (indicated by the color of the star in Figure 4c) and the location of the fluid front (color contours). This enables the reader to compare the event time and the location of the front. There is a clear sequence of times from early to late events. Overlapping events from different grains was not a significant factor because crushing of the grains during assembly resulted in a shorter particle lifetime (See Figure S6(left) for information on initial grain area versus duration of AE.).*

Another major concern I have, is how can the authors confirm that as the dust particles decrease in size (disaggregate and dissolve) they do not become mobilized and travel at or behind the fluid-air interface?

R3.17 *This is a good question. The digital imaging is not at a sufficiently high enough resolution to image particle transport. Given the distribution of events in Figure 4c and the error in location, local transport may occur. We now note this in the manuscript:*

“Some of the locations of the emissions are near the original placement of the dust while others are not. During set-up of the fracture, the dust is crushed when the two surfaces are registered resulting in multiple particles (e.g. multiple stars near sensors 1, 5, and 6 in Figure 4b) that can potentially move with the invading front. The acoustic locations agree with the visual location of the fluid front to within ± 5 mm. This agreement illustrates the potential to use chattering dust to track fluid-front movement through fractures in opaque material (i.e. rock) at the laboratory scale and potentially at the near bore hole scale. The location uncertainty is caused by the water-air interface, the fracture roughness, and the high angle detection geometry.”

Why did the authors cut the bottom of the aperture field in Figure 4(b) from that used by Su et al. (1999) this seems misleading.

R3.18 *This was originally done because the sensors were only located on a portion of the sample. We re-did the experiments to include imaging of the fluid front over the entire fracture plane and now show the full aperture distribution (Figure 4a).*

Experiment 2: Gravity-driven motion of a bluff particle in a fracture of constant aperture

Dissolving or disaggregating particle: My major concern is that we have not been shown how a single particle behaves in the calibration section. While figure 2 does show some relationship between bubbles and emissions this is not as important as the single dust particle size evolution versus time (Figure 3 shows how an amalgamation of dust particles behaves). Obviously, in the context of the Laplace-Young theory presented, particle size and aperture are key parameters (also fluid viscosity, which likely changes over time or near the particles boundary layer as the sucrose changes from a solid to aqueous state but this is likely a second-order effect). Without this knowledge of the particles true

behavior it is difficult to claim this has no effect.

R3.19 As mentioned in **R3.10**, grain characterization has now been included in the Supplemental Methods. All dust starts out as a “single” grain. Given the structure of the dust (gas bubbles in a dissolvable matrix), whether a “single” grain disaggregates or remains intact depends on the thickness of the matrix between gas bubbles, the volume of the gas bubbles, and the location of the bubbles. As shown in Figure S2 and Figures 1 & 2 in the manuscript, some single grains contain fewer larger bubble volumes than others.

We now include bubble dissolution information for 12 twelve grains (DT04-DT14) that were tested and identified which samples experienced disaggregation in the Supplemental Methods (section S4 & see response to reviewer **R3.10**). Graphs are also included in this section on the duration of AE emissions, time rate of change of grain cross-sectional area and change in number of hits per time. (Note: The behavior of a grain that disaggregated before settling under gravity in a fracture is shown in Figure S9c in Section 6.3).

Complex source: I have drawn a schematic representation of what general seeded fault might look like for experiment 1. The panel on the left shows that it appears that the intact dust particle must be crushed to fit in the aperture in the seeded fault experiment. In the right panel, if the fractured practice emits a explosive source and the grain is in contact with the wall, this might be different that if the particle has dissolved and no longer touches the solid fracture wall. Either way, the manner in which the acoustics are interpreted are treated very poorly in this study and I think a more rigorous calibration study is merited before conclusion are drawn.

Experiment 1

R3.20 The location method is based on differences in arrival times among the sensors. For the uniform aperture fracture, this method involved finding the arrival time of the first maximum from the Hilbert transformation of the signals (see Figure S7 and section S6.3). We did explore the effect of refraction on the error in dust location during descent and descent speed, $V_{descent}$ using a forward modeling approach (see response to reviewer **R3.22**). Improving location methods with forward and/or inverse modeling is an ongoing research area in both the AE and seismology communities, and will be an area for future work for interpreting dust location in more complicated systems which is beyond the scope of this paper.

In contrast to the “seeded” particles in experiment 1, the other experiments (2,3,4 and 5) have their own wave propagation problems (detailed schematically below).

While the Broyden Method is referred to in the Methods as the manner in which they solve the linear set of equations to determine the location, they do not take into account that the source might be more complicated than they propose.

R3.21 *We now include the source characteristics for a dust grain in an unconfined fluid (see Section S3 and Figure S5). The source is explosive. When the source is confined in a rough fracture, the explosive signatures still remain in that the 1st motion at each sensor shows the same excursion in amplitude (see Figure below which is only in this response to reviewer not in the paper or supplemental methods). As mentioned above, because the arrival time is the time when the transducer was triggered, the characteristics of the source do not affect interpretation of the arrival time. In general, location errors are more likely to arise from anisotropy in velocities (not relevant for the samples used in these studies), high detection angles, wave-guiding, heterogeneous material velocity, etc. all of which are common sources of error in non-destructive testing as well as in earthquake seismology.*

From their logic, the particle falls due to gravity and is suspended in a fluid. The chemical reaction with the dust emits a source (or hit) which is therefore a suspended "point source" in a fluid. The pressure waves will then move through the fluid and interact with the fracture wall. Wave will then be produced in the acrylic blocks but assuming that the waves are always traveling through the acrylic solid (as they did when assuming a fault plane) is incorrect. You can also see on the inset images of Figure 6, the yellow orange dots that depict the path of descent (from the video) is not planar. This complex fluid structure interaction encountered by the elastodynamic waves needs more through calibration in a separate study.

Below is a schematic depiction of the two problems that the authors have not appropriately calibrated or considered in this study.

Experiments 2, 3, 4 and 5

The level of detail provide into the acoustic data processing in Figure 9 and the type of signals observed are not well-described. I would suspect that a more thorough study that described in a more rigorous manner how acoustic data looks and more details to each experiment would make the sources of errors and problems I mentioned above more.

R3.22 *We performed and now include a theoretical study (Sections S6.3 & S6.4) to examine the effect of refraction on dust location within a fluid-filled aperture on the interpretation of the location of the emission. Using Fermat's principle of path of least time, a forward modeling study was performed to determine the error in location and error in the interpretation of the descent speed of the dust assuming a point source. The arrival time from the source to all 8 transducers (using the layout for the uniform-aperture fracture Table S7 and Figure S8) were calculated for the path of least time (refracted path) and the direct path (no refraction). Then the values were used with the Broyden solver to determine the locations for the two methods.*

Here is the information from the Supplemental Methods:

“The difference in travel time between the two ray paths is shown in Figure S10right for all simulated and measured apertures. For apertures $b = 0.5$ to 2 mm, the time difference is below the resolution of AE monitoring system. The AE system records signals with $0.5 \mu\text{s}/\text{point}$. A $dt = 0.5 \mu\text{s}$ in acrylic yields a difference in path length of 1.365 mm. For $b > 2$ mm, the difference in arrival time increases.

The travel times between each source position and 8 sensor locations (4 on the $z=-0.75$ x-y plane and 4 on the $z=+0.75$ mm x-y plane with the locations used in the experiments, see Table S7 & Figure S8b) were calculated to determine the error in x location and interpretation of the speed of descent of a dust particle. The time differences (assuming sensor 1 as the reference as in the data analysis of the experiments) were used in the Broyden non-linear solver described above to determine the location of the source as it fell along the path $(x,z)=(0,0)$ for $y=-0.75$ to $+0.75$ mm in 1 mm increments for apertures 0.5, 1, 2, 4, 8, and 10 mm.

As in the experiments, a fitted velocity, fitted- $V_{acrylic}$, was determined that yielded a total travel path of the dust ~ 150 mm. For a grain falling along the $(x,z)=(0,0)$, Tables S8 lists the average and standard deviation in the x location of the dust and the ratio of the speed of descent based on including refraction, V_{dr} , to the speed of descent assuming a direct path, V_{dd} , between the source and receiver, as assumed in the experiments by fitting $V_{acrylic} = 2717$ m/s for all apertures. The x location deviated from $x=0$ by $157 \mu\text{m}$ to 2.6 mm with increasing aperture and resulted in 1-3% under prediction of the speed of descent, $V_{descent}$ (Table S8). When $V_{acrylic}$ is interpreted for each aperture (Table S9), the under-prediction in $V_{descent}$

decreases and the interpreted x location is off from $x=0$ by 100 μm to 550 μm with increasing aperture.

In general, deviations in the interpretation of dust location and speed of descent increase with increasing aperture. For the experiments presented here, the error from assuming a direct path in the interpretation of speed of descent is on the order of 1-3% which is smaller than the error from experimental measurements from averaging the location from sensor groups for each event.

Table S8. From simulations, the ratio of the speed of descent when including refraction compared to the direct path, average interpreted x location and standard deviation for a range of apertures assuming $V_{acrylic} = 2717$ m/s for all apertures.

Aperture (mm)	V_{dr}/V_{dd}	Average x location (mm)	STD in x (mm)
0.5	1.009	0.157	0.184
1	1.010	0.275	0.155
2	1.013	0.544	0.146
4	1.018	1.069	0.262
8	1.028	2.102	0.590
10	1.03	2.601	0.756

Table S9. From simulations, the ratio of the speed of descent when including refraction compared to the direct path, average interpreted x location and standard deviation for a range of apertures by fitting $V_{acrylic}$ for each aperture. “

Aperture (mm)	Fitted $V_{acrylic}$ (m/s)	V_{dr}/V_{dd}	Average x location (mm)	STD in x (mm)
0.5	2715	1.010	0.285	0.204
1	2717	1.010	0.275	0.158
2	2720	1.011	0.352	0.100
4	2728	1.011	0.370	0.176
8	2743	1.011	0.501	0.505
10	2751	1.012	0.550	0.657

From reviewer 3: Please see the Annotated PDF for minor comments.

C.2 Annotated Comments (copied from pdf)

Number 1. .

Number 2.

MAJOR POINT:

The methodology surrounding the XRCT scanning is not clearly presented or sufficiently described to make this experiment repeatable by others. Please provide the setting, scanning apparatus, resolution of the scans. As it stands, this is not an acceptable description of the methods. Please consult Pini and Madonna (2016, <https://link.springer.com/article/10.1007/s10934-015-0085-8>). They discuss the different techniques required to use a medical X-ray CT (mCT) scanning instrument (General Electric hi-speed CT/i X-ray computed tomography) or the Synchrotron-based X-ray radiation (CT). As they note, the types of CT will vary and how the data is processed will be machine- and user-dependent. Please also consult comments on figure 1.

EDIT: I found that the authors have referenced the scanners model in the acknowledgments: "...Purdue

University for acquisition of the Zeiss Xradia 510 3D X-ray Microscope". This still does not sufficiently describe the methods used to describe how the authors have measured the sizes of CO₂ pores within the sucrose grain.

I have looked up the maximum resolution Zeiss Xradia 510 3D X-ray Microscope

(<https://applications.zeiss.com/>

C125792900358A3F/0/90AEFFB81C61FBC5C1257BC0003B5276/\$FILE/EN_40_011_007_xradia-510-versa_rell1-1.pdf)

See response R3.1

According to the data sheet, the spatial resolution can range from 0.7 microns to 70 nanometer. This will undoubtedly impact the smallest visible bubble size (3 microns diameter) and therefore needs to be mentioned?

See response R3.1

Number 3. it is unclear if these dimensions are what is shown in the Figure 1(b). Perhaps showing the full scan of the sample and a visual image of the dust will be more informative than what is shown currently in this image. I do not feel that the 2D projection is very illustrative, and if this is a portion of the scan that was used for the tomographic reconstruction, can the authors show in Figure 1(b) where this projection plane is?

See response R3.2

Number 4. Is it a single grain? If so the caption does not confirm this mentioning: "gas bubbles in reactive grains". Please clarify this point.

See response R3.3

The dissemination between a single and multiple grains are important since in Figure 2 the authors are showing the average acoustic behavior of single grains.

See responses R3.3c & R3.10

Number 5. Figure 3 shows the dust grain to be larger than ~ 1 mm. At 10 seconds, the grain is almost on the order of the scale = 4.3 mm). The caption states that this is multiple grains but this is here it says this is a single dust grain. Please clarify.

Please also see responses on Figure 3.

See responses R3.3c & R3.10

Number 6 While this is not part of the main experiment, the "calibration" and understanding the "acoustic-bubble size" relationship is crucial to the explanation of the experiments later. I do not feel convinced that the authors have described this sufficiently for other to reproduce these results.

See responses R3.1, R3.2, R3.3b & R3.4

Number 7 Please see notes on Figure 3

See responses R3.5

Number 8 I believe that there are other factors that will control fluid intrusion. Dynamic viscosity, fluid density, pressure gradients, flow rate and micro-mechanical contact forces also have an influence on the fluid invasion process

See responses R3.12

Number 9 Laplace

We left this as Laplace-Young's equation because we are referring to

$$\begin{aligned}\Delta p &= -\gamma \nabla \cdot \hat{n} \\ &= -\gamma H_f \\ &= -\gamma \left(\frac{1}{R_1} + \frac{1}{R_2} \right)\end{aligned}$$

(from Wikipedia site for Laplace-Young's Equation) where the capillary pressure is proportional to the curvature of the fluid-fluid interface which is related to the size of the aperture or pore.

Number 10 It is unclear how the authors "seeded" the fault, this will have direct effects on the density of emissions .

See responses R3.17

Number 11. If the objective is to track the penetration front and if the fracture was a transparent replica, the position of the interface is visible. How does the AE records of the fluid-penetration match with the observations? Moreover, the authors again do not describe the experimental specifics. Simply mentioning that they were similar to Su et al. (1999) is not adequate. If they look at the experiments have important parameters such as the specific inclination angle and inlet flow rates, which are not mentioned here. In its current state this seeding experiment is unacceptable for publication.

See responses R3.15, R3.16

Details on the experiment are now provided in the Supplemental Methods.

Page 3

Number 1: The fracture aperture is ~ 5 to 27% the average size of of a single grain of dust. Are the authors compressing and fracturing dust particles along the interface?

See responses R3.17 & Supplemental Methods Section S5.1

Fig 3 in Su et al. (1999) shows probability distribution function from the acrylic analogs. It shows that virtually none of the aperture measurements are larger than the mean Dust particle (~1mm).

Number 2 Su et al. (1999) performed a range of experiments they detailed appropriately. This is not the case here

See response R3.13

Number 3 There is no basis to this claim. They have not shown concomitant experimental results for acoustic hits and locations and the fluid penetration front. I believe this claim is hearsay.

It is also unclear as to how the front was determined. Based on the calibration curves shown in Figure 3, regions where water has penetrated initially will be continuously emitting quite frequently for the first 100 seconds.

See responses R3.14, R3.15 & R3.16

Again, the methods do not seem to accurately account for the fact the dust will continue to emit at locations where the water has already penetrated. The authors note that the red circles are locations of the events but neglect to mention how they separate the newly activated Dust particles at the fluid front from the previously activated particles using the raw acoustic records.

I do not believe the "seeded" fault experiment is sufficiently described by the authors.

See response R3.14, R3.15 & R3.16

Number 4 Why is Figure 5b given before 5a?

Figures have been re-numbered and some removed.

Number 5

Q1: How accurate are hydrodynamics only models at this scale? Are there electrical charges influencing the particle settling.

We have no method for measuring any electrical charges on the dust or in the water.

Q2: The methods do not accurately describe the experiments. There is not indication of the amount of Dust used and, referring back to the "seeded" dust experiment, the acoustic emission and how the front is separated from background source "noise" is not clear in the least.

The dataset shown in Figure 9 shows an acoustic emission event but it is unclear what this means. Is there also likely background noise events if the dust behaves as describe by the calibration experiment in Figure 3. The authors should describe how they disseminate this in the to retrieve the measurements in Figures 4a, 5, 6, 7b and 8.

Thresholds were set in the AE measurements to minimize "noise" (i.e. electrical). Noise from the invading fluid front (with no chattering dust) occurred in the seeded experiments if the threshold was set as low as 25 dB. By using a threshold of 40 dB, no "noise" emissions were received at 4 or more transducers until the front reached/passed a dust grain (see Figure 4 in the manuscript). The "new" Supplement Methods document describes our experimental and data analysis approach.

Number 6 Was this due to surface tension and the clumping of the agglomeration of particles? It is confusing since the authors appear to show a clump of grains in Figure 3 and don't truly define what a single grains decay function looks like.

Please see the comments related to disaggregation (ex. **R3.3c & R3.10**) and the Supplemental Methods (Section S4, Figures S6, Section S6.3 and Figure S9).

Judging from the calibration curves and photos in figure 4, the particles will have disaggregated and dissolved. The theory by Goldman et al. (1967), takes into account a bluff body that has a constant aspect ratio.

However, we know that the particles are shrinking over time form Figure 3. It does not appear that the authors take this into account.

This is now discussed in Supplement Methods Section 6.3.

Page 4

Number 1. Considering the mass of particles in the fluid, is this a Newtonian fluid or fluid with dispersed a large concentration of interacting spheres, which will change th effective viscosity of the fluid (Einstein's expression with $(1+2.5 \times \text{volume fraction})$) ?

The fluid for all cases was water which is Newtonian under ambient conditions. For calibration experiments, cells were refreshed with clean water between measurements. For the uniform- and variable-aperture fracture experiments, the water was refreshed every time the aperture was changed. The seeded fracture experiments used fresh water every time. The quantity of dissolved sucrose was not of sufficient concentration to change the viscosity of the fluid.

Number 2. My major concerns detailed up to this point hold true from the results shown in the variable aperture fractures.

See all previous comments.

Page 5.

Number 1 This is a major concern in that the authors have not appreciably calibrated the dusts behavior in this regard.

We have now included an extensive new Supplement Method document. We thank the reviewer for his detailed reading of the manuscript which enabled us to compose the Supplemental Method to contain the information most readers would want to know.

Number 2. This is a major concern in that the authors have not appreciably calibrated how dust geometrically disaggregates and dissolves over time. Since we see through Stokes-Laplace theorem where they will settle depends on particle size, without a fixed understanding of how particle size degrades over time there will be no relationship to any fracture properties.

We now discuss the effects of disaggregation on dissolution rates and functional form (S4, and show an example of a disaggregated grain behavior when sampling and uniform aperture fracture (Figure S9c Section S6.3)

Page 7.

Number 1 What velocity, specify.

This is now specified in the Supplemental Methods document in Section 6.3 Interpretation of Chattering Dust Location from Acoustic Emission. Specifically:

c) a system velocity of $V=2630$ m/s where the system includes the acrylic blocks and the water-saturated fracture and is based on the arrival time of the peak of the Hilbert which is later than the first break.

Page 8

Number 1. Link and article not found

The reference now includes the link and the date the link was viewed.

“On May 13, 2020 the article was on the following url: <https://explorer.aapg.org/story?articleid=126>.”

Page 11

Number 1-5 related to figures

Number 6. This large dominant inclusion does not appear in the Figure 1(b) which leads me to think that Figure 1(a) is not used in the 3D reconstruction. This is fine but it does illuminate the point that there is a wide range of distribution of CO₂ pores sizes.

See Responses R3.3b & R3.6

In Figure 3(a) the authors choose only to use the "average bubble size". I think that a standard deviation on this metric should also be described in the graphic (see note on figure 3). Perhaps the authors should show the probability distribution function (PDF) as a part of Figure 1.

Moreover, the authors need to clarify what "bubble size" means. Visually, it appears that the bubble form spheres and they later use an approximate diameter to describe the yellow bubble. I believe that the authors should be able to describe the PDF. While this point may seem moot, I note that
ENERGY = PRESSURE x VOLUME.

If we assume that all pore contain CO₂ pressurized at ~4.1 MPa, then the energy released will scale as
ENERGY \propto DIAMETER³.

Where the energy will determine the intensity of the recorded seismicity (assuming constant seismic efficiency, Aki and Richards, 2002 https://www.ideo.columbia.edu/~richards/Aki_Richards.html).

See Response R3.8

Number 7-11. Related to figures

The figure has been replaced with new images and data.

Number 12. Is this picture of a single grain or multiple grains? Please clarify and be consistent with the text.

See Response R3.3a

Number 13. Please place axes on both images. It will help the reader if the 3D edges to the scan is given. As mentioned above. The methodology surrounding the scan is not well described in the body of the text or the "methods" section.

See Response R3.2

Number 14-19 Related to figures

Figures have been re-drafted or removed.

Number 20. Please see note on describing the PDF of bubble sizes above. EDIT: Based on my remarks above. It is important to determine the average hits versus bubble size to see if larger detections are attributed to larger average pores since there appears to be no clear trend -- and the authors do not comment on a trend -- between the hits and number of bubbles or average bubble size.

See Response R3.2, R3.3b

Page 12.

Number 1. X-axis:

Q1. Number of signals (/sec?)

Q2. Are "signals" equivalent to AE hits from Figure 2?

We now define "hit" in the Supplemental methods.

The axes have been changed to read "Number of Hits per Second".

Number 2.

Major Comment:

Firstly please see the additional commented on the lack of methods describing the how to calculate the decay of the area fraction with time. The authors mention that from the initial amalgamate of Dust particles the particles will disaggregate upon the introduction of water. This can be clearly seen between the image at 10 and 100s. Due to the mobility issues associated with the flow of particles within a fracture, how these particles penetrate the fault will be dependent on how they disaggregate and their particle size evolution over time. In the current state, the area fraction does not provide sufficient understanding of the Dust particles/amalgamates The authors should be able to quantify how disaggregation occurs using relatively straightforward image detection algorithm in MATLAB (see regionprops Selvadurai, 2015; Gonzalez et al., 2009). Perhaps the number of particles and average particle area would be a more interesting due geometric constraint place by the fracture aperture. (Gonzalez, R.C.; Woods, R.; Eddins, S.L. Digital Image Processing Using MATLAB; Gatesmark Publishing: Knoxville, TN, USA, 2009.)

We have added an extensive supplement methods documents to provide the experimental and analysis

details. See Responses R3.10 & Figure 3.

Number 3.

Exponential decay of acoustic hits emissions with disaggregation and dissolution of dust particle over time:

It appears that the dust response empirically fits the following equation:

$$N = 150 \exp(-0.015*t)$$

where N = hits/second. Integrating this we see that approximately ~10,000 hits were recorded over this experiment. From their original claims that grains are ~ 1 mm in length scale and produce ~ 100s to 1000s events. Can the authors please clarify exactly how many grains were originally placed in the chamber?

See Response R3.10 & Figure 3.

Number 4.

Y-axis methods:

I did not find anywhere in the "methods" the manner in which the Area fraction was calculated. Again, as is the case with Figures 1 and 2, there is insufficient description of the metrics used here that are supposed to calibrate the Dust particles.

I will guess that they used some type of image detection algorithms.

All image analysis methods used for the dissolution experiments, the settling under gravity experiments and seeded fracture experiments can be found in supplemental methods sections S4, S6.2 and S5.2, respectively.

Page 13

Number: 1 Author: Reviewer Subject: Callout Date: 26/11/2019 12:58:18 PM

According to the authors the average particle size of a dust particle is at this level.

See response R3.13 & R3.14.

Number: 2 Author: Reviewer Subject: Line Date: 26/11/2019 12:58:28 PM

See response R3.13 & R3.14.

Number: 3 Author: Reviewer Subject: Text Box Date: 26/11/2019 4:38:15 PM

????????????????????

This figure has been re-drafted and a scale bar included in the image.

Number: 4 Author: Reviewer Subject: Sticky Note Date: 26/11/2019 4:39:06 PM

Can the authors please comment as to why the full aperture map from Su et al. (1999) is not shown? This is misleading.

See Response R3.18.

Number: 5 Author: Reviewer Subject: Sticky Note Date: 27/11/2019 6:23:41 AM

It may be that these boundaries are the artifact of the contour scheme. They have chosen 7 contours to show a much more complex process. The scatter points of the locations should from their first emission should be

colored with respect to time. The current state of the plot might be misleading.

The experiments were repeated with full digital imaging of the fracture plane (Figure 4 in the manuscript). The contours are now from image analysis of the digital images taken during the fluid invasion process.

Number: 6 Author: Reviewer Subject: Sticky Note Date: 27/11/2019 6:23:54 AM

Can the authors confirm that the particles are not moving with the fluid flow. One would suspect that as the mobility of seeded dust particles increase, i.e. as the mass of the seed decreases, the particles might move with the front.

See Response R3.17.

Number: 7 Author: Reviewer Subject: Sticky Note Date: 27/11/2019 6:24:26 AM

Please place scale (x- & y-) dimensions on the aperture map in (b).

A scale is now at the bottom of the figure and the dimensions of the sample (as measured in our laboratory) are given in the Supplemental Methods S5.

Number: 8 Author: Reviewer Subject: Sticky Note Date: 26/11/2019 2:33:40 PM

Is there a similar amount of dust as in Figure 3 deployed?

Single dust grains were used in the experiments with dimensions of 1-6 mm as measured by eye with a ruler. The particles are ellipsoidal in nature and thus the range of dimensions.

Number: 9 Author: Reviewer Subject: Sticky Note Date: 26/11/2019 2:49:58 PM

MAJOR COMMENT: While the Broyden Method is referred to in the Methods as the manner in which they solve the linear set of equations to determine the location, they do not take into account that the source might be more complicated than they propose. From their logic, the particle falls and while doing so it is suspended in the fluid. The chemical reaction with the dust emits a source (or hit) which is therefore a suspended "point source" in a fluid. The pressure waves will then move through the fluid and interact with the fracture wall. Wave will then be produced in the acrylic blocks but assuming that the waves are always traveling through the acrylic solid (as they did when assuming a fault plane) is incorrect. You can also see on the inset images of Figure 6, the yellow orange dots that depict the path of descent (from the video) is not planar.

See Response R3.22.

Number: 10 Author: Reviewer Subject: Sticky Note Date: 26/11/2019 2:33:37 PM Can the authors discuss the error bars and why they appear to increase then the particles are in a known location.

The following explanation has been placed in the supplemental methods document section 6.3.

“This approach yielded the full possible descent path from 0 to 150 mm depth. Once the velocity was determined, events were located by using a Broyden non-linear solver to fit the x & y positions of the source and the time to source for sets of transducer combination (in groups of 3). The error bars shown in Figure 5a in the manuscript are based on the averaged location from all of the groups for a given event. The error bars in Figure 5b in the manuscript are from the average of 7-10 tests performed for each aperture.”

When a grain is falling, it is better situated with respect to the transducer locations.

Number: 11 Author: Reviewer Subject: Sticky Note Date: 26/11/2019 2:31:18 PM Can the authors explain comment on how the non-linear decrease in particle size (from Figure 3) will affect the measured speed?

In section S6.3 of the supplemental methods we discuss the effect of area reduction during the time of descent.

D. Reviewer #4

Report on: “Probing Complex Geophysical Geometries with Chattering Dust” submitted to Nature Communication

General comments to Authors: The manuscript describes application of geophysical acoustic methods –acoustic emissions-to track chemically-activated “chattering” dust particles as they flow through fractures. Authors claim that with monitoring the acoustic emission’s source locations they could infer aperture distribution of fractures or interfaces.

The core idea is to track the source location of “chattering” particles while they go through fracture. In the following I will argue the method might not be accurate and there are some flaws in the described method and employed techniques.

R4.1 *Pertaining to the question of accuracy, the location method is based on differences in arrival times among the sensors. For the uniform aperture fracture, this method involved finding the arrival time of the first maximum from the Hilbert transformation of the signals (see Figure S7 and section S6.3). For the other experiments, the locations were calculated by the AE system software (AEWin) which bases arrival-time differences on the time of the test when a sensor is triggered. Thus there is no interpretation of phase of the signal in our analysis. We did explore the effect of refraction on the error in dust location during descent and descent speed, V_{descent} using a forward modeling approach (see response to reviewer R3.22). In general, deviations in the interpretation of dust location and speed of descent increase with increasing aperture. For the experiments presented here, the error from assuming a direct path in the interpretation of speed of descent is on the order of 1-3% which is smaller than the error from experimental measurements from averaging the location from sensor groups for each event. Improving location methods with forward and/or inverse modeling is an ongoing research area in both the AE and seismology communities, and will be an area for future work for interpreting dust location in more complicated systems which is beyond the scope of this paper.*

1- how dissolving reactive grains accurately interact with the fracture topography’s complexity – considering that pressure carbon dioxide (4.1Mpa) and the coating quality which suppose to dissolve? How this is dependent on the speed of fluid flow and interaction/hydro- mechanical interaction of a grain or collective of grains with the fracture surface?

R4.2 *We agree that dissolution rates depend on hydrodynamic conditions as well as the particle shape and size. The dissolution rate as a function of time was measured to be linear for a single grain (see sample DT4 in Figure 4 in the manuscript) indicating that dissolution rates are fairly constant for 1 grain in a quiescent fluid. Nonlinear behavior in dissolution rate was observed for particles that disaggregated. When a grain breaks into multiple grains, more surface area is created for dissolution (see sample DT14 in Figure 4).*

In the seeded experiment where fluid was slowly imbibed into a fracture, the grains were observed to emit over a period of 1-2 minutes. Initial grain sizes were 3-5 mm in diameter prior to crushing. After assembling the fracture (see response to reviewer 3: R3.13 & R3.17), the observable grains had post-

crushing diameters < 2 mm. The 1-2 minute duration of acoustic emission is consistent with the duration of emission based on initial particle size (Figure S6(left) in the supplemental methods). For flow through the intersecting fracture using 3-5 mm diameter grains, acoustic emissions occurred over a 2-3 minute period. For the experiments in this manuscript, the flow rates did not have a significant effect on the dissolution rates.

2- Another important point which I believe authors ignored is to discuss the accuracy of their source locations as well as characterization of amplifier-sensor response to see how signal is distorted or does have phase shift.

R4.3 *In response to the reviewer's comment, the source characteristics are now discussed in for dust in an unconfined fluid (see Section S3 and Figure S5 in the Supplemental Methods). The source is explosive (see Figure S5). Whether in an unconfined fluid or confined in a rough fracture, the explosive signatures still remain in that the 1st motion at each sensor shows the same excursion in amplitude (see the figure in response to reviewer 3 **R3.21** & Figure S5). Therefore, the phase characteristics of the source do not affect interpretation of the arrival time. In general, location errors arise from the same problems faced by the nondestructive testing and seismology communities, such as anisotropy in velocities (not relevant for the samples used in these studies), high detection angles, wave-guiding, heterogeneous material velocity, etc.*

3-The acoustic emission sources could be very different ranging from structural phase transition to different types of defects. Here as I am sure most of the events are from explosion of grains but the time line of explosion which occurs in the most converging point of the fractures (?) would restrict the application of the method. In fact, a single grain might generate many events and in this point it is not clear to me how a single grain (precise chemical or mechanical formulation is needed for this) emits acoustic phonons; the rate of emissions as well as their characteristics have not been studied and it is not clear how one can distinguish these signal from other corrosion or flow induced signals

R4.5 *Examples of the emitted signals are shown in Figures S3, S5 and in this response to reviewers (see **R3.21**). We now state in the manuscript:*

“As it dissolves (at a rate of $\sim 0.115 \text{ mm}^2/\text{s} \pm 0.014 \text{ mm}^2/\text{s}$ see Figure 3), the average time between recorded emissions is approximately 30 milliseconds and the average amplitudes of the events remain relatively constant during dissolution (Figure 3c&f).”

As only first arrivals are used that fall within the maximum time window (i.e. maximum distance between a transducer and any point on the fracture plane), overlapping signals have not been an issue for these experiments with single grains. Also, thresholds were set in the AE measurements to minimize “noise” (i.e. electrical). Noise from the invading fluid front (with no chattering dust) occurred in the seeded experiments if the threshold was set as low as 25 dB. But by using a threshold of 40 dB, no signals were received at 4 or more transducers until the front reached/passed a dust grain (see Figure 4 in the manuscript). No corrosion of the samples was observed, and the transducers for the seeded, uniform- and variable-aperture fractures were never in contact with the fluid. Water-coupled transducers were used in the “T” intersecting fractures experiments but these sensors are designed for submersion periods of 12 hours. The experiments on the “T” fractures took less than 10 minutes per trial and only 10-15 trials per day.

Granted that these laboratory experiments are idealized, but AE signals from water invading fractures or pores in rock are typically less than 40 dB and hence would not masquerade as dust events.

4-Apart of the above points (and many other points), I do not see any fundamental or intriguing point

in this paper regarding both natural emissions or study of complexity of fracture topography (or aperture map).

R4.6 *A long standing challenge in subsurface and laboratory science is to characterize fractures remotely in opaque rock or civilian infrastructure. Although current active monitoring methods detect the presence of fractures, fracture properties (e.g. fracture specific stiffness) are inferred from wave velocities, amplitudes and/or spectral properties. Similarly, passive monitoring methods rely on natural or induced seismicity (or acoustic emission) which are then interpreted to provide information on location of fractures but provide no information on fracture properties. What neither of these techniques can currently provide is the connectivity among fractures, i.e. how fluid may flow through a fracture network. We agree that the method presented in this manuscript is not ready for field scale usage (see discussion related to reviewer 2: **R2.1**), but this method is completely applicable to laboratory and infrastructure-scale studies. Chattering dust provides a non-destructive method for tracking flow paths, and it can be performed repeatably as a fractured rock sample is subjected to stress or variations in fluid pressure or other physical processes.*

Reviewers' comments:

Reviewer #2 (Remarks to the Author):

Major Claims: Our energy and environmental systems rely on the flow of fluids through fractures. Roughly 85% of our energy still is from subsurface extraction of fossil fuels making fracture flow in subsurface systems a very impactful, broad topic of interest. However, fracture flow in the subsurface cannot be imaged directly since these operations generally take place thousands of feet below the ground making it difficult to optimize or control fractured systems. This manuscript employs an innovative approach to utilize "chattering dust" to track chemically-activated dust particles that emit acoustic emissions as they dissolve and flow through a fracture network. The key advantage of the approach is that it has the potential to delineate the transport path through a fracture network which would be transformational for optimizing and eventually controlling fracture flow in subsurface systems. It is well understood that fluids only access a small percentage of fractured rocks and we have very little knowledge of which fractures actually flow in subsurface systems. The approach combines ideas from tracer testing and remote geophysical monitoring to illuminate connected flow paths through a fracture system.

Conclusions: The paper demonstrates a compelling proof of concept of using chattering dust to determine flow paths through a fracture network in the laboratory. The authors state that a key question of the approach refers to scalability of the approach. I would like to see more discussion on scalability since it is key to the eventual utility of the approach in the field. Detecting chattering dust at the lab scale versus trying to detect acoustic emissions that are generated thousands of feet below the ground in noisy environments is a very different challenge. For example, oil and gas companies have been trying proppants that emit acoustic emissions to determine how far proppants transport into a fracture system to determine their effectiveness. So far these methods have been ineffective in the field.

It is true that we currently have no reliable methods to illuminate 3D flow paths even at the lab scale through opaque rock samples. I would like more discussion on why knowing 3D flow paths even at the lab scale pushes the field forward since moving to the field scale seems to be a big leap. I believe illuminating 3D flow paths at the lab scale is already an impactful result since apertures that can be measured at the lab scale (micron-mm apertures) can conduct enough flow to be relevant in unconventional oil and gas production during late production times or in leading to leakage from carbon sequestration sites that rely on low permeability caprocks to contain CO₂. If moving to the field is not a big leap, then this work is very impactful. I'd like more discussion on how this technique could eventually be deployed in the field. It is ok even if multiple steps need to be taken and proven to eventually get there. The authors do provide numbers on the magnitude of the emissions and what is detectable. Therefore, expanding this discussion would be of interest to readers.

Methods: In general, I found the method technically sound and well explained. As far as the technique, the chattering dust is always changing as it flows and dissolves through a fracture network. It seems like this would make interpreting the results of these experiments more challenging. Is the current interpretation technique of the acoustic emissions result in a unique delineation of the flow path through the system? Also due to the complex nature of the chattering dust, are the experiments reproducible? Some comments on reproducibility should be provided.

In summary, I found this to be an innovative paper on an impactful topic that I believe should be published in Nature Communications if my comments are addressed. I believe the paper would be greatly strengthened if more of the advantages and drawbacks of the methods at various scales were spelled out more clearly. If this technique is limited to the lab scale, is this enough of success?

Reviewer #3 (Remarks to the Author):

Please see the two attached PDF:

General comments (Major comments): [FINAL] Reviewer_Comments.pdf

Annotated PDF (major and minor comments): [FINAL] Annotated Comments.pdf

Reviewer #4 (Remarks to the Author):

Report on: "Probing Complex Geophysical Geometries with Chattering Dust" submitted to Nature Communication

General comments to Authors: The manuscript describes application of geophysical acoustic methods –acoustic emissions-to track chemically-activated "chattering" dust particles as they flow through fractures. Authors claim that with monitoring the acoustic emission's source locations they could infer aperture distribution of fractures or interfaces.

The core idea is to track the source location of "chattering" particles while they go through fracture. In the following I will argue the method might not be accurate and there are some flaws in the described method and employed techniques.

1-how dissolving reactive grains accurately interact with the fracture topography's complexity – considering that pressure carbon dioxide (4.1Mpa) and the coating quality which suppose to dissolve? How this is dependent on the speed of fluid flow and interaction/hydro- mechanical interaction of a grain or collective of grains with the fracture surface?

2- Another important point which I believe authors ignored is to discuss the accuracy of their source locations as well as characterization of amplifier-sensor response to see how signal is distorted or does have phase shift.

The acoustic emission sources could be very different ranging from structural phase transition to different types of defects. Here as I am sure most of the events are from explosion of grains but the time line of explosion which occurs in the most converging point of the fractures (?) would restrict the application of the method. In fact, a single grain might generate many events and in this point it is not clear to me how a single grain (precise chemical or mechanical formulation is needed for this) emits acoustic phonons; the rate of emissions as well as their characteristics have not been studied and it is not clear how one can distinguish these signal from other corrosion or flow induced signals

Apart of the above points (and many other points), I do not see any fundamental or intriguing point in this paper regarding both natural emissions or study of complexity of fracture topography (or aperture map).

Reviewer's comments to the Authors

In the manuscript entitled: "Probing Complex Geophysical Geometries with Chattering Dust" by Pyrak-Nolte et al., the authors attempt to deploy so-called "chattering dust" in five different experimental configurations. In these experiments this novel acoustic source is deployed in an attempt to probe properties of fluid transport along analog experimental faults. These types of studies are very important to both geophysical and geomechanical fields of studies. Some quantities they attempt to quantify is the rate of fluid intrusion into a dry, rough analog fracture, the particle velocities of particles driven by gravity in idealized fault with increasingly complex cross-sectional profiles, flow of fluids in matrix-fracture interfaces and complex fluid flow fields occurring at an "inverted-T" fracture joint.

Principle of the novel sensor: Chattering dust is a novel organic acoustic source composed of a sucrose-based body filled with many (1000s to 10,000s) spherical inclusions of pressurized CO₂ gas ($P_{pore} \sim 4.1$ MPa). The authors' showed that when the a Dust particle comes in contact with water, the sucrose body dissolves, likely converted to an aqueous solution, and, in certain cases, the pressurized pores suddenly discharge energy as the pore pressure equalizes to the ambient surrounding. This sudden discharge makes an explosive acoustic (fluid) source, or it "chatters", within the fluid. This will cause pressure waves to propagate (discussed later as a major comment) that travel through a fracture, interacts with the wall and results, then propagate through the solid material govern by the equations of motions. Between 8-15 passive acoustic emission sensors are used to locate the sources (chatter) using the first (body wave) arrivals and a basic triangulation approach common in the field of laboratory acoustic emission studies.

I commend the authors on this version of the manuscript. The level of detail supplied in the supplements has greatly increased my ability to discern the experiments and extract valuable information. I would like to thank the authors for addressing the numerous question, comments and concerns I made with their initial submission. Apart from some small typographic errors (see annotated PDFs), *I believe this manuscript is acceptable for publication in Nature Communications.*

Minor comments:

1. **Please provide the mark, brand or product number for the actual chattering dust.** The authors must give state that it is a commercial product but have omitted this detail.
2. The authors' discuss an energy metric ($E = PV$) in the Results (Chattering dust) section but then use an acoustic amplitude in dB. I suspect they are using the conversion of voltage to dB from the PZT transducers. This was confusing. Please clarify how you are calculating the amplitude in units of dB. In the discussion, they mention that an explosive event could emit $1 \mu\text{J}$ of energy. I assume this is determined from the explosive source ($P = 4.1$ MPa) associated with a spherical inclusion with radius $r = 38\mu\text{m}$. Could the author's please reference of include the calculation from how to go from scalar seismic moment $M_0 = 1 \mu\text{J} = 1 \mu\text{Nm}$ to moment magnitude $M_w = (2/3) (\log_{10}(M_0) - 9.05) = -10$ (Kanamori & Hanks, 1975).

Please see the Annotated PDF for minor comments.

Sincerely,

Paul A. Selvadurai

Report on the revised version of “*Probing Complex Geophysical Geometries with Chattering Dust*”

I went through one more time the article and proposal which authors put together. While the paper improved in many ways however, I think the proposed method will not be practical or needs a lot of approximations in implementing/using in natural scales. Apart of this point which needs a long-term plan and collaboration with some oil-industry groups, I particularly red the supplementary materials and the details of explanations. Clearly the major focus is to accurately estimate source location of events while the focus is correlation the bubble (dust) dissolution /explosion events to geometry of the fractures. I will list some points which in my opinion - whom spent last 10 years on developing sensors/algorithms on piezo-elements/AEs and investigate also on complexity of interfaces – need smore work and explanation rather superficial mentioning them.

- 1- Source locations are based on Hilbert envelope rather than first arrival time – I do not understand why this is the case -the peak of the Hilbert envelope is a few microsecond delays with first arrival time – What are the effects of glue-interfaces in source location ?
- 2- It is still a question how many events a static dust and dynamic dust create considering a fixed distance of the bubble from sensor-array ? What is the minimum energy which one exploding bubble crate and can be detected with calibrated piezo-sensors ? Authors might be able to estimate this in meV and relate the energy of waves using some calibration constants to piezo-elements- This is important to scale up the results to some realistic set-up in natural scales – Authors miss the proper calculations on relating emitting energy from 1)single particle 2) collective particles to weak energy seismic signals
- 3- The main problem, in my opinion is that the third particles (dusts) as moving emitting acoustic phonon particles are not emitting waves in a controllable way – clearly they are not tuned perfectly to “probe” uniformly the topology of the fracture system – either in a simple geometry ,the probe signal is not uniform. I would assume Figure S1 is an attempt to relate numbers of bubbles in a grain to emitted signals but

is not complete and need more works. As homework, authors could estimate how (if) an exploding bubble within packed of bubble can vibrate the grain and neighborhood bubbles /grain and then a solid-interface.

- 4- I understand the authors use just one aspect of the array record of AEs (i.e., source location) to sort of map the geometry/topology of channels - a simple discussion for source locations in a natural set-up must use many simplification (earth model /layers/unknown dissipation sources etc) – How do we scale up these results ?a major question left without any touch on that –
- 5- How temperature will effect these results – a new story I guess which clearly is the case for real case studies beneath the earth

While I found the work generally interesting but in comparing with other advances in studies of acoustic emissions -from basic physics, shedding further lights on importance of resolution of fraction of angstrom of displacements in piezo-elements positions and relating them to basic sources of crackling emissions- has trivial significant.

Response to the Reviewers

Manuscript: “Probing Complex Geophysical Geometries with Chattering Dust”

Authors: Pyrak-Nolte et al.

The authors thank all of the reviewers for their comments and thoughtful questions. Below are our responses to the comments on the revised manuscript and the supplemental information document. The discussion to the paper has been rewritten to address the comments related to moving the method from the laboratory to larger-scale applications, and a new section of the supplemental information (S9) contains measurements made at distances up to 2 meters on concrete (Figure S11). The updated discussion is highlighted in yellow in the manuscript file.

Reviewer 2

Reviewer #2 (Remarks to the Author):

Major Claims: This manuscript employs an innovative approach to utilize “chattering dust” to track chemically-activated dust particles that emit acoustic emissions as they dissolve and flow through a fracture network. The key advantage of the approach is that it has the potential to delineate the transport path through a fracture network which would be transformational for optimizing and eventually controlling fracture flow in subsurface systems. It is well understood that fluids only access a small percentage of fractured rocks and we have very little knowledge of which fractures actually flow in fractured systems which occur in opaque media. The approach combines ideas from tracer testing and remote geophysical monitoring to illuminate connected flow paths through a fracture system. My main concerns on the major claims for the manuscript have been addressed by the revision. Specifically, the authors have explained the benefits of using this technique at the laboratory scale is already a major step forward since it will illuminate the flow paths in opaque materials such as rock. These lab scale studies will provide insight for applications such as dams, bridges, and near wellbore monitoring where small scale cracks may eventually lead to large catastrophic cracks. The manuscript now describes the challenges for deployment to the field scale. These seem to be big challenges to extend to larger scale subsurface applications but I am satisfied that even at the lab scale that this is a large step forward for imaging fractures through opaque media. The revision does a much better job in providing quantitative numbers on the relevant scales, detection limits, etc. The strengths and weaknesses of the approach are now clearly explained.

Conclusions: The paper demonstrates a compelling proof of concept of using chattering dust to determine flow paths through a fracture network in the laboratory. In the revision, the authors now clearly address the challenges for scalability of the approach. The paper also motivates the laboratory work better showing that illuminating flow paths in fractures at the lab scale through opaque samples is an important step forward for multiple applications.

Methods: In general, I found the method technically sound and well explained in the original submission. However, the revision greatly improves the methods with the supplemental information. One concern I had was that the chattering dust is always changing as it flows and dissolves through a fracture network. It seems like this would make interpreting the results of these experiments more challenging. It was not clear how repeatable the experiments could be. The supplemental information rigorously addresses the question of repeatability. In addition it explains that repeatability is a function of dust grain volume, strength of interpretation method, as well as disaggregation of a particle. This does seem a complex set of parameters that would affect repeatability. However, imaging flow paths in opaque material is a difficult endeavor and I’m satisfied with how the authors have described issues with repeatability.

In summary, I found this to be an innovative paper on an impactful topic that was greatly improved after the revisions. The supplemental information does a good job of quantifying the methods and their

repeatability for readers that want more information. I believe this improved revision should be published in Nature Communications.

Reviewer 3

In the manuscript entitled: “Probing Complex Geophysical Geometries with Chattering Dust” by Pyrak-Nolte et al., the authors attempt to deploy so-called “chattering dust” in five different experimental configurations. In these experiments this novel acoustic source is deployed in an attempt to probe properties of fluid transport along analog experimental faults. These types of studies are very important to both geophysical and geomechanical fields of studies. Some quantities they attempt to quantify is the rate of fluid intrusion into a dry, rough analog fracture, the particle velocities of particles driven by gravity in idealized fault with increasingly complex cross-sectional profiles, flow of fluids in matrix-fracture interfaces and complex fluid flow fields occurring at an “inverted-T” fracture joint.

Principle of the novel sensor: Chattering dust is a novel organic acoustic source composed of a sucrose-based body filled with many (1000s to 10,000s) spherical inclusions of pressurized CO₂ gas (P_{pore} ~ 4.1 MPa). The authors’ showed that when the a Dust particle comes in contact with water, the sucrose body dissolves, likely converted to an aqueous solution, and, in certain cases, the pressurized pores suddenly discharge energy as the pore pressure equalizes to the ambient surrounding. This sudden discharge makes an explosive acoustic (fluid) source, or it “chatters”, within the fluid. This will cause pressure waves to propagate (discussed later as a major comment) that travel though a fracture, interacts with the wall and results, then propagate through the solid material govern by the equations of motions. Between 8-15 passive acoustic emission sensors are used to locate the sources (chatter) using the first (body wave) arrivals and a basic triangulation approach common in the field of laboratory acoustic emission studies.

I commend the authors on this version of the manuscript. The level of detail supplied in the supplements has greatly increased my ability to discern the experiments and extract valuable information. I would like to thank the authors for addressing the numerous question, comments and concerns I made with their initial submission. Apart from some small typographic errors (see annotated PDFs), I believe this manuscript is acceptable for publication in Nature Communications.

1. 1. Please provide the mark, brand or product number for the actual chattering dust. The authors must give state that it is a commercial product but have omitted this detail.

We have added the following information in the supplemental information S1 subsection S1.1:

“(made by Zeta Espacial, S.A.)”

2. The authors’ discuss an energy metric ($E = PV$) in the Results (Chattering dust) section but then use an acoustic amplitude in dB. I suspect they are using the conversion of voltage to dB from the PZT transducers. This was confusing. Please clarify how you are calculating the amplitude in units of dB. In the discussion, they mention that an explosive event could emit 1 μJ of energy. I assume this is determined from the explosive source ($P = 4.1 \text{ MPa}$) associated with a spherical inclusion with radius $r = 38\mu\text{m}$. Could the author’s please reference of include the calculation from how to go from scalar seismic moment $M_0 = 1 \mu\text{J} = 1 \mu\text{Nm}$ to moment magnitude $M_w = (2/3) (\log_{10}(M_0) - 9.05) = -10$ (Kanamori & Hanks, 1975).

In section S2 we note

“The AE software converts Volts to dB: $dB = 20 \log(V_{\text{max}}/1 \mu\text{Volt}) - (\text{Preamplifier Gain in dB}).$ ”

We removed the following statement from the Supplemental Information

~~“An estimate of the energy released can be obtained from the amplitude, as the energy of a signal is proportional to the amplitude squared.”~~

In the manuscript, we added:

“The received signal in units of dB is proportional to the \log_{10} of the received energy.”

and we moved the sentence in the manuscript: *“The energy of release, E , is proportional to the volume, V , of gas ($E = PV$ where P is pressure).”* to the discussion.

We added an additional section in the supplemental methods to support the quoted magnitude of -10.

“S8. Calculation of Moment Magnitude

The energy, E , released from an average bubble ($r=28 \mu\text{m}$) with a pressure 4.1 MPa is roughly $E \sim 0.4 \mu\text{J}$. Assuming $E \sim M_o$, where M_o is the seismic moment in Nm, the estimated moment magnitude, M_w^4 , for a single bubble determined from $M_w = (\log_{10} M_o - 9.05)/1.5$ is $M_w \sim -10$.”

**Hanks, T. C. and H. Kanamori, A moment magnitude scale, Journal of Geophysical Research, vol. 84, no B5, 2348-2350 (1979).*

Comments from Annotated Manuscript

Please describe the sections.

Could they just refer the reader to these figures (in main text and supplemental) at later sections of the results? I think this will improve the manuscript.

We removed or reduced the number of references to the Figures to make the paragraph more readable:

“Chattering dust is composed of commercially available reactive grains made of sucrose containing pockets of pressurized carbon dioxide (4.1 MPa) (Figures 1 & 2). As the coating dissolves, the compressed gas is explosively released, yielding acoustic emissions (e.g. Figures S4 & S6c). X-ray microscopy shows that the dust grains contain spherical bubbles of pressurized gas that range in size from 4.5 to 270 micrometers (Figures 1 & 2). For instance, a single reactive dust grain contains a distribution of bubble sizes throughout the bulk of the sample. The volume probability distribution functions for the bubbles compared to the distribution functions for the amplitudes of the events show relatively similar functional forms (Figure 2).”

Section?

We added the section number (S4) in for the supplemental method.

Perhaps I think this is till a “proof of concept” experiment. Maybe I do not understand how you would “pre-seed” the fracture. (Which is why you do the next experiments, I presume.) Could you elaborate on this statement?

We added the following to the manuscript:

“However, seeding a fracture in the field would be complicated. Seeding of a near borehole fracture could be performed by pumping the dust in a nonreactive fluid (e.g. low viscosity oil or dry gas) into a fracture, followed by an invasion of a reactive fluid (water) to release the emissions. “

Figure not in order.

The figures have been re-ordered.

What is the cause of the spread in this? It is much larger than the AE location error. Is it just different spatial distinct voids in the grain reacting?

As observed in the error analysis (S6.4), maximum error in travel time differences occur when the dust is far away from a receiver or at high angle detection. In the location approach, events were selected if they were recorded by 3 or more AE sensors. No filter was used to eliminate sensors based on distance from the source. When the dust is floating on the top of the fracture, the dust is far away from the 4 lower transducers. Future applications could optimize the transducer layout and use of additional sensors to reduce the spread.

Comments from Annotated Supplemental Information (check our consistency on how I refer to this)

This section is very well done and has addressed my initial concerns about experimental repeatability. The authors have gone through exceptional lengths and provided more than adequate detail to the experiments. Please note check carefully for spelling and typos.

We have fixed the following comments and typos highlighted by the reviewer:

Page 1: 4.1 Pa to 4.1 MPa

Page 4: wave form to waveform

Page 6: Were the sensors suppose to appear hear? This figure is to show the coordinate system used for these experiments and the authors felt a 3D drawing with the transducers would not be clear. The transducers locations are provided in Table S3 for this coordinate system.

Page 9: wave form signal to waveform signal

Page 10: wave form signal to waveform signal

Page 13: V_{descen} to $V_{descent}$

Page 15: $y = -0.75$ to $+ 0.75$ mm to $y = -0.075$ to 0.075 mm.

Reviewer 4

I went through one more time the article and proposal which authors put together. While the paper improved in many ways however, I think the proposed method will not be practical or needs a lot of approximations in implementing/using in natural scales. Apart of this point which needs a long-term plan and collaboration with some oil-industry groups, I particularly red the supplementary materials and the details of explanations. Clearly the major focus is to accurately estimate source location of events while the focus is correlation the bubble (dust) dissolution /explosion events to geometry of the fractures. I will list some points which in my opinion -whom spent last 10 years on developing sensors/algorithms on piezo-elements/AEs and investigate also on complexity of interfaces – need smore work and explanation rather superficial mentioning them.

1- Source locations are based on Hilbert envelope rather than first arrival time – I do not understand why this is the case -the peak of the Hilbert envelope is a few microsecond delays with first arrival time – What are the effects of glue-interfaces in source location ?

Analyses were first performed using the 1st arrival for location, and then performed using the time of the Hilbert peak. Using the peak of the Hilbert transform for interpretation of arrival times for the dust in the uniform aperture fractures is more stable. The time offset is compensated for by using a modified velocity.

As stated in the supplemental information section S6.3, “...a system velocity of $V=2630$ m/s where the system includes the acrylic blocks and the water-saturated fracture and is based on the arrival time of the peak of the Hilbert which is later than the first break. The acoustic speed in acrylic is 2730 m/s and for water at 20°C the speed is 1480 m/s. The selected value of V was based on a minimization approach that takes advantage of the dust floating on the surface of the water prior to descending into the fracture. While the dust is floating, it provides a reference or calibration point for determining the velocity of the system.”

The speed of sound in the hot glue is ~1870 m/s (This speed has been added to the supplemental information in S6.3). All transducers were coupled in a similar manner with ~ 1mm layer. Because only time differences are used in the location analysis, the additional delay caused by the couplant is removed.

2- It is still a question how many events a static dust and dynamic dust create considering a fixed distance of the bubble from sensor-array ? What is the minimum energy which one exploding bubble create and can be detected with calibrated piezo-sensors ? Authors might be able to estimate this in mV and relate the energy of waves using some calibration constants to piezo-elements- This is important to scale up the results to some realistic set-up in natural scales – Authors miss the proper calculations on relating emitting energy from 1)single particle 2) collective particles to weak energy seismic signals

From Figure S1 & Figure 2, the smallest bubble size that is detectable near a transducer is estimated to have a radius of 10 micrometers and an emitted energy of roughly 1 nJ. This is estimated from the signal-to-noise ratio of the transducers and the comparison of the detected amplitude with the bubble volume distribution. For application to a more extended system, such as civil infrastructure monitoring, sensors may be a meter from the source and would require an energy of 100 nJ. A bubble volume with this energy is approximately at the peak in the bubble distribution. Recent developments with distributed acoustic sensing (DAS) have achieved 40 nJ detection.

We now include Supplement Information Section S9 that demonstrates the change in signal with distance from a transducer out to 2 meters.

We now include the calculation of moment magnitude in Supplement Information Section S8 to provide a reference frame for the magnitude of dust events relative to other seismic events.

3- The main problem, in my opinion is that the third particles (dusts) as moving emitting acoustic phonon particles are not emitting waves in a controllable way –clearly they are not tuned perfectly to “probe” uniformly the topology of the fracture system – either in a simple geometry ,the probe signal is not uniform. I would assume Figure S1 is an attempt to relate numbers of bubbles in a grain to emitted signals but is not complete and need more works. As homework, authors could estimate how (if) an exploding bubble within packed of bubble can vibrate the grain and neighborhood

bubbles /grain and then a solid-interface

The fields of acoustic emissions, earthquake seismology and induced seismicity are based on interpreting signals from uncontrollable sources that are not “tuned” for the task at hand, can vary in phase, and may only emit a small number of events, all of which can vary in amplitude and frequency. While the number of bubbles (and in turn the number of events) and the amplitude of the signals from the dust vary among grains, the amplitude of the signal does not affect the arrival times. Being an explosive source, any observed changes in phase at the receiver arise from the material the waves propagated through. The dust provides a better source than random crack events in the earth or in materials.

The events are released from the surface of the dust adjacent to the water so there is no intervening material. The dust is smaller than the detection wavelength so it acts as a point source.

4- I understand the authors use just one aspect of the array record of AEs (i.e., source location) to sort of map the geometry/topology of channels - a simple discussion for source locations in a natural set-up must use many simplification (earth model /layers/unknown dissipation sources etc) – How do we scale up these results ?a major question left without any touch on that –

In the manuscript, the last paragraph of the discussion included a discussion on scaling from the laboratory to the field scale which it made it clear that there are hurdles to overcome but that applications to civilian infrastructure or near borehole (1m) are quite doable. This supported by the data in Figure S11 in the Supplement Information Section S9 that shows dust generated signals as a function of distance from a transducer out to 2 meters. The discussion has been rewritten.

5- How temperature will effect these results – a new story I guess which clearly is the case for real case studies beneath the earth

Of course temperature will effect how rapidly the gas expands upon dissolution of the dust. This falls under the statement from the discussion “...will require additional research to improve the life time of the dust (which may require new chemical formulations)”.

Reviewer #4 (Remarks to the Author):

Report on the revised version of the manuscript-

I read the revised version of the work and authors responses to my critics and other referees. While I appreciate for the answers and other referees positive feed-backs, but I can not find this work a distinguished work - also the answers of the authors to some of my questions are not satisfactory - for instance :I asked to develop a simple multi-grain model with minimum numerical simulation of wave propagation and contrarian the limitation of the method and etc - Most importantly , the presented work for me doe snot add anything to concept of ultrasound developments or novel understanding of the material behavior in terms of angstrom resolution of kHz long wavelength waveform - The second point which I could not satisfy myself with the description presented in the paper is how these results scales up not only in terms of simplified energy density argument of AEs(Aki's frequency*length) to the natural cases but other complexities arise in that case. No doubt that works is interesting and valid in many points but it is more presentable with more details in Geophysics journals - there is no significant discovery or big/major steps towards multi-aspects of topology of fracture networks and their characteristics with some moving bubbles -

Response to the Reviewers

Manuscript: “Probing Complex Geophysical Geometries with Chattering Dust”

Authors: Pyrak-Nolte et al.

REVIEWER’S COMMENTS

Reviewer #4 (Remarks to the Author):

Report on the revised version of the manuscript-

“I red the revised version of the work and authors responses to my critics and other referees. While I appreciate for the answers and other referees positive feed-backs, but I can not find this work a distinguished work - also the answers of the authors to some of my questions are not satisfactory - for instance :I asked to develop a simple multi-grain model with minimum numerical simulation of wave propagation and contrarian the limitation of the method and etc - Most importantly , the presented work for me doe snot add anything to concept of ultrasound developments or novel understanding of the material behavior in terms of angstrom resolution of kHz long wavelength waveform - The second point which I could not satisfy myself with the description presented in the paper is how these results scales up not only in terms of simplified energy density argument of AEs(Aki's frequency*length) to the natural cases but other complexities arise in that case. No doubt that works is interesting and valid in many points but it is more presentable with more details in Geophysics journals - there is no significant discovery or big/major steps towards multi-aspects of topology of fracture networks and their characteristics with some moving bubbles –“

Reviewer’s Comment: “:I asked to develop a simple multi-grain model with minimum numerical simulation of wave propagation and contrarian the limitation of the method and etc –“

In a previous review, the reviewer had indeed asked “As homework, authors could estimate how (if) an exploding bubble within packed of bubble can vibrate the grain and neighborhood bubbles /grain and then a solid-interface.”

Response: Numerical simulation is not trivial, it is beyond the scope of the current work, and such a simulation would not affect the measurements nor conclusions in this study. Shock wave modeling of this problem requires inclusion of chemically reactive dissolution rates, evolving fluid-fluid and fluid-solid interfacial tensions, evolving fluid & solid properties, cavitation and other complex multiphase physics flows (i.e. jetting). In addition, the simulations would need appropriate time steps to resolve all time and length scales. Numerical simulations of a single non-spherical bubble collapsing without grains nearby was the topic of a Ph.D thesis (Johnsen, 2007, UC Berkeley). Therefore, this request is beyond the scope of this paper.

Reviewer's Comment: “Most importantly , the presented work for me doe snot add anything to concept of ultrasound developments or novel understanding of the material behavior in terms of angstrom resolution of kHz long wavelength waveform –“

Response: This work is not about ultrasonic developments or material behavior—it is about a fundamentally new approach to interrogating, remotely, fracture connectivity, inferring changes in fracture aperture and flow paths through a fracture or a system of fractures. No other approach currently exists that can do this. No claims are made in the paper about angstrom resolution from kHz waveforms. All information about fracture connectivity and fracture aperture arise from interpretation of acoustic emission location and how that location changes as a function of time.

Reviewer's Comment: The second point which I could not satisfy myself with the description presented in the paper is how these results scales up not only in terms of simplified energy density argument of AEs(Aki's frequency*length) to the natural cases but other complexities arise in that case.

Response: The reviewer states that he is not satisfied but does provide specific concerns that we have not already addressed. Another reviewer asked us to place the energy released from the dust in the context of current standards for natural and induced seismicity that are used to interpret subsurface behavior. We feel this addressed the other reviewer's comments relevant for the AE and geophysics community.

Reviewer's Comment: No doubt that works is interesting and valid in many points but it is more presentable with more details in Geophysics journals - there is no significant discovery or big/major steps towards multi-aspects of topology of fracture networks and their characteristics with some moving bubbles –

Response: As stated in our response above, no other approach currently exists to track fracture connectivity in real time. Our method has the ability to place sources directly inside fractures which is not possible even with fabricated MEMS devices which jam fractures when the device is larger than the aperture. With a dissolving source, this limit is overcome and, as shown in the manuscript (Figure 3), even as the dust dissolves and decreases in size the amplitudes remain relatively constant. In addition, the dust is environmentally friendly compared to fabricated devices. Imagine if this could be adapted for use in the human body to provide a new imaging modality without the need for patient to pass devices through their bodies. The applicability of this approach is only limited by the imagination of the reader.